# Exceptional stability of a perilipin on lipid droplets depends on its polar residues, suggesting multimeric assembly

Manuel Giménez-Andrés[1,2], Tadej Emeršič[3], Sandra Antoine-Bally[1], Juan Martin D'Ambrosio[1,4], Bruno Antonny[5], Jure Derganc[3,6], Alenka Čopič[1,4]*

[1]Institut Jacques Monod, CNRS, Université de Paris, Paris, France; [2]Université Paris-Saclay, Saint-Aubin, France; [3]Institute of Biophysics, Faculty of Medicine, University of Ljubljana, Ljubljana, Slovenia; [4]CRBM, University of Montpellier and CNRS, Montpellier, France; [5]Université Côte d'Azur, CNRS, IPMC, Valbonne, France; [6]Chair of Microprocess Engineering and Technology – COMPETE, University of Ljubljana, Ljubljana, Slovenia

**Abstract** Numerous proteins target lipid droplets (LDs) through amphipathic helices (AHs). It is generally assumed that AHs insert bulky hydrophobic residues in packing defects at the LD surface. However, this model does not explain the targeting of perilipins, the most abundant and specific amphipathic proteins of LDs, which are weakly hydrophobic. A striking example is Plin4, whose gigantic and repetitive AH lacks bulky hydrophobic residues. Using a range of complementary approaches, we show that Plin4 forms a remarkably immobile and stable protein layer at the surface of cellular or in vitro generated oil droplets, and decreases LD size. Plin4 AH stability on LDs is exquisitely sensitive to the nature and distribution of its polar residues. These results suggest that Plin4 forms stable arrangements of adjacent AHs via polar/electrostatic interactions, reminiscent of the organization of apolipoproteins in lipoprotein particles, thus pointing to a general mechanism of AH stabilization via lateral interactions.

*For correspondence:
alenka.copic@crbm.cnrs.fr

**Competing interests:** The authors declare that no competing interests exist.

## Introduction

Lipid droplets (LDs) are cellular organelles specialized for storage of lipids and maintenance of cellular lipid homeostasis. They are composed of a neutral lipid core that is covered by a monolayer of phospholipids and other amphiphilic lipids, and by proteins (*Thiam et al., 2013b*; *Olzmann and Carvalho, 2019*). LDs vary in size over four orders of magnitude, depending on organism/cell type and fasting state of a cell; in mature adipocytes, the majority of the cell can be occupied by a single LD measuring >100 μm in diameter (*Lundquist et al., 2020*; *Stenkula and Erlanson-Albertsson, 2018*). Secreted lipoprotein particles are similar to LDs in terms of their over-all composition but are much smaller, ranging from 10 to 1000 nm in diameter (*Ohsaki et al., 2014*).

Different types of proteins have been found to associate with LDs (*Brasaemle et al., 2004*; *Bersuker et al., 2018*; *Kory et al., 2015*; *Pataki et al., 2018*; *Mejhert et al., 2020*). They can be either stably embedded in the LD monolayer, coming by diffusion from the endoplasmic reticulum (ER), from which LDs emerge, or they associate with LDs peripherally from the cytosol (*Ohsaki et al., 2014*; *Bersuker and Olzmann, 2017*). Many of them are enzymes involved in lipid synthesis or hydrolysis, for example triglyceride synthases, acyltransferases, lipases and their inhibitors or activators (*Wilfling et al., 2013*; *Zechner et al., 2017*). Proteins can also regulate LDs in a non-enzymatic manner. A prominent example is the perilipins: in mammals, this is a family of five proteins that share related structural features and are highly abundant on LDs (*Sztalryd and Brasaemle, 2017*). They vary in their tissue distribution: Plin2 and Plin3 are widely expressed, whereas Plin1 and Plin4 are

most highly expressed in adipocytes, and Plin5 is enriched in oxidative tissues (*Wolins et al., 2006*; *Brasaemle et al., 2004*; *Wolins et al., 2003*). Plin4 and Plin5 are also enriched in muscle tissues. Less closely related abundant LD proteins have been identified in many other species (*Gao et al., 2017*; *Granneman et al., 2017*; *Miura et al., 2002*). Whereas perilipins contain no known enzymatic motifs, a number of them, in particular Plin1, have been shown to regulate the recruitment of lipases to the LD surface (*Sztalryd and Brasaemle, 2017*).

Many LD-localized proteins use amphipathic helices (AHs) to directly interact with the LD lipid surface (*Bersuker and Olzmann, 2017*; *Giménez-Andrés et al., 2018*). All mammalian perilipins contain a predicted AH region in their N-terminal part, which has been shown to be important for their LD localization (*McManaman et al., 2003*; *Nakamura and Fujimoto, 2003*; *Bulankina et al., 2009*; *Rowe et al., 2016*; *Čopič et al., 2018*). This region is composed of 11-aa repeats that would fold into a 3–11 helix, a slightly extended variant of an α-helix that was characterized in α-synuclein and in apolipoproteins (*Bussell and Eliezer, 2003*; *Jao et al., 2008*). Other regions, including a C-terminal region that can fold into a four-helix bundle (*Hickenbottom et al., 2004*), can also contribute to LD targeting to varying extents (*Targett-Adams et al., 2003*; *Subramanian et al., 2004*; *Nakamura et al., 2004*; *Ajjaji et al., 2019*). Interestingly, both the 11-aa repeat region and the four-helix bundle bear structural similarities with apolipoproteins, which are required for formation of lipoprotein particles (*Saito et al., 2003*; *Melchior et al., 2017*).

The 11-aa repeat AH region is by far the longest in Plin4, containing close to 1000 aa in the human protein, with repeats that are highly homologous at the 33-aa level (*Čopič et al., 2018*; *Scherer et al., 1998*). The aa composition of Plin4 AH reveals a striking bias toward small hydrophobic residues, in particular V, T, and A, whereas large residues such as W and F are almost entirely absent (*Figure 1A*). We have demonstrated that the Plin4 AH region is unfolded in solution, but adopts a highly helical structure in contact with a lipid surface. The low hydrophobicity of this AH promotes specific targeting to LDs, which are permissive for the binding of many amphipathic proteins (*Čopič et al., 2018*; *Prévost et al., 2018*). This is likely due to the physical properties of the LD surface, where the spreading of the phospholipid monolayer leads to exposure of the hydrophobic core with which the hydrophobic face of an AH can interact more strongly (*Bacle et al., 2017*; *Chorlay et al., 2019*). Due to its extreme length, Plin4 in particular can cover a large LD surface and could act as a substitute for phospholipids (*Čopič et al., 2018*). A recent study has identified expansion of Plin4 33-aa repeats in a family with a rare autosomal-dominant progressive myopathy, underscoring the importance of studying this protein (*Ruggieri et al., 2020*).

Due to their high abundance on LDs, perilipins are often referred to as LD coat proteins (*Sztalryd and Brasaemle, 2017*). Protein coats have been well characterized on transport vesicles, for example COPI, COPII, and clathrin coat. In all these cases, the coat forms in a tightly controlled manner by sequential recruitment of coat subunits on the membrane surface (*Schekman and Orci, 1996*; *Taylor et al., 2011*). Importantly, coat subunits laterally interact to form a highly polymerized structure covering the surface of a vesicle (*Faini et al., 2013*). Coat polymerization is in fact the main force that generates these membrane vesicles (*Saleem et al., 2015*). Perilipins have not been shown to be directly involved in LD budding from the ER; LD formation may be principally driven by lipids, with proteins playing a more regulatory role (*Ben M'barek et al., 2017*; *Chorlay et al., 2019*; *Santinho et al., 2020*). On the other hand, COPI coat components have also been observed to bind to and influence LDs and to regulate recruitment of other LD proteins (*Guo et al., 2008*; *Thiam et al., 2013a*; *Wilfling et al., 2014*; *Soni et al., 2009*).

Here, we ask whether perilipins possess any of the qualities traditionally associated with protein coats. We focus on the 11-aa repeat AH regions of mammalian perilipins, which directly associate with the lipid surface of LDs. We analyze the stability of perilipin AHs on the lipid surface and their ability to form an immobile structure using various cellular and biochemical approaches, as well as a novel microfluidics set-up to follow the interaction of AHs with oil over time. We show that one perilipin, Plin4, is capable of making highly stable protein-lipid structures by forming an immobile coat on the surface of pure oil or LDs in cells using its unique AH. The Plin4-oil droplets remain stable over the course of many days. In contrast, the interaction of the AHs from other perilipins with LDs or with oil is highly dynamic. The stability of interaction correlates with the size of oil particles, which are smaller in the case of Plin4. Extensive mutagenesis shows that the AH of Plin4 can form an immobile coat due to its organized structure that could enable interhelical interactions on the lipid

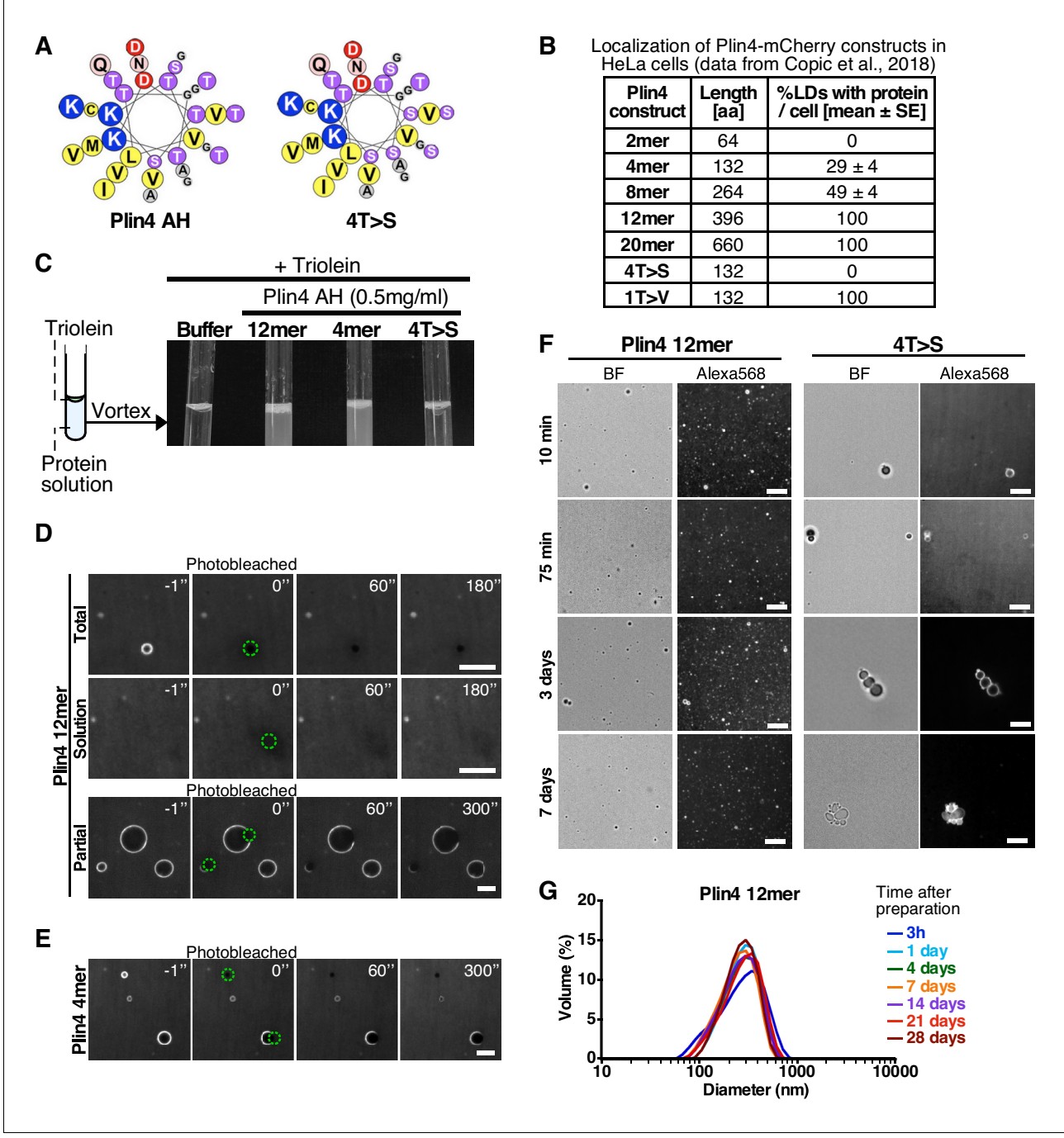

**Figure 1.** Plin4 AH forms very stable oil particles. (**A**) Helical wheel representations of Plin4 and Plin4 4T > S AHs. (**B**) Summary of LD localization of Plin4 AH and mutants (*Čopič et al., 2018*). (**C**) A 10 μl drop of triolein was added to 190 μl of HK buffer containing the indicated proteins (Plin4 12mer, Plin4 4mer or 4T > S, all at 0.5 mg/ml). After vigorous vortexing, the samples were photographed. (**D**) Dynamics of Plin4 12mer interaction with oil as assessed by FRAP assays. Emulsions of triolein with unlabeled Plin4 12mer (0.5 mg/ml) and Alexa488-labeled Plin4 12mer (0.025 mg/ml) 12mer were prepared as in C and visualized by fluorescence microscopy. FRAP was performed on large droplets, which were entirely bleached (top row), or in the bulk as a control (middle row). The lower row shows a FRAP experiment performed on a limited region of the droplet. A summary of all FRAP experiments is shown in *Figure 1—figure supplement 1B*. (**E**) FRAP of Plin4 4mer (0.5 mg/ml) mixed with Alexa488-labeled Plin4 4mer (0.01 mg/ml). (**F**) Light microscopy images of Plin4 12mer and 4T > S emulsions at time points after preparation by vortexing. (**G**) Size distribution as assessed by dynamic light scattering (DLS) of a Plin4 12mer/triolein emulsion from 3 hr to 28 days after the vortexing reaction. Experiment was repeated two times. Scale bars: 5 μm.

The online version of this article includes the following figure supplement(s) for figure 1:

**Figure supplement 1.** Summary of FRAP experiments on oil droplets.

surface. This model is supported by our measurement of Plin4 density, which reveals tight packing of helices on oil surface.

## Results

### Purified Plin4 AH forms very stable protein-oil emulsions

We have previously shown that the AH of Plin4 is optimized for LD binding both by its length and particular aa composition (*Čopič et al., 2018*). The AH sequence of human Plin4 is composed of ~29 highly homologous 33-aa repeats (*Figure 1A*). The efficiency of LD targeting correlated with AH length: at least four 33-aa repeats of the wild-type Plin4 AH were needed to detect some LD localization in HeLa cells (*Figure 1B*). Furthermore, the strong bias toward small residues is decisive for LD targeting: mutations that increased hydrophobicity (T > V) made Plin4 promiscuous for other organelles besides LDs; mutations that decreased hydrophobicity (T > S) rendered Plin4 cytosolic (*Figure 1B*).

The amphipathic region of Plin4 is capable of emulsifying triolein upon vigorous mechanical mixing (vortex) in aqueous buffer in the absence of any other surfactant, such as phospholipids. A similar result was obtained with a shorter Plin4 AH construct containing 4 33-aa repeats (Plin4 4mer) or a longer construct comprising 12 33-aa repeats (Plin4 12mer) (*Figure 1C*; *Čopič et al., 2018*). Electron microscopy and dynamic light scattering (DLS) revealed that emulsions of triolein and Plin4 12mer contained spherical oil particles with a large range of sizes; typically with a diameter of 50 to 500 nm (mean ≈ 200 nm), although some larger particles (diameter >1 μm) could also be observed (*Čopič et al., 2018*). We focused on these latter particles, whose size made them suitable for imaging by fluorescence light microscopy. For this, we performed triolein emulsification in the presence Plin4 12mer or Plin4 4mer labeled with the fluorescent dye Alexa488 (Plin4 12mer-A488), mixed with unlabeled Plin4 AH of the same length. The spherical particles displayed a homogenous fluorescent surface, which allowed us to perform dynamics measurements by fluorescence recovery after photobleaching (FRAP) (*Figure 1D,E*).

In the first FRAP protocol, we bleached an entire Plin4-oil particle. In this case, fluorescence recovery should occur by exchange between free Plin4 AH-A488 in solution and bleached Plin4 AH-A488 molecules bound to the lipoparticle surface. As shown in *Figure 1D,E*, we detected no recovery within the time range of the measurement (3 min). In the second FRAP protocol, we bleached a limited area of the lipoparticle surface to follow fluorescence recovery of Plin4 AH molecules by lateral diffusion. Again, we observed no recovery within 5 min after bleaching, neither with Plin4 4mer nor Plin4 12mer (*Figure 1D,E*). Thus, Plin4 AH forms a very stable and immobile coat at the surface of triolein.

To gain further insight into the stability of the Plin4 AH/triolein particles, we visualized the emulsions prepared with fluorescent Plin4 12mer over 7 days after the vortexing step. As shown in *Figure 1F*, the emulsions at t = 15 min, 75 min, 4 days and 7 days appeared very similar, showing numerous submicrometer particles and a few larger (>1 μm) particles. Importantly, most if not all particles remained isolated during this long observation time, showing no obvious clustering or aggregation.

For comparison, we used a mutated form of Plin4 4mer (4T > S), in which several threonines in the hydrophobic face of the amphipathic region had been replaced by the more polar residue serine (*Figure 1A*). Plin4(4T > S) was purified from bacteria following the same procedure as for Plin4 4mer or Plin4 12mer. This mutant was inefficient at emulsifying olein (*Figure 1C*), in agreement with its inability to target LDs in HeLa cells (*Figure 1B*; *Čopič et al., 2018*). However, we could observe a few large triolein droplets formed by fluorescently labeled Plin4(4T > S). The lipoparticles covered by Plin4(4T > S) clustered over time, suggesting that their coat was much less protective than that observed with Plin4 12mer (*Figure 1F*). FRAP analysis of these particles revealed a variable behavior of Plin4(4T > S), in agreement with its decreased emulsifying capacity and the tendency to cluster; some clustering could already be detected in some of the FRAP recovery time-courses (*Figure 1— figure supplement 1*).

Analysis by DLS revealed an even more remarkable stability of Plin4 12mer-oil particles over time, as we detected no change in particle size distribution even 28 days after emulsification (*Figure 1G*). This puts Plin4 AH on par with natural emulsifiers used for technological purposes in food or

pharmaceutical industry (*McClements and Gumus, 2016*). In contrast, the particles formed by Plin4 (4T > S) were too heterogenous for analysis by DLS even at the first time-point (3 hr) after emulsion formation.

## Following Plin4-oil interaction in real time using microfluidics

To further study the interaction of Plin4 AH with neutral lipids, we required a method where we could present the AH to the oil surface in a gentle manner and follow in real time the assembly of protein on the oil surface. We developed a microfluidic system, in which we used a glass chip with two channels joined by a T-junction. We introduced the water-based buffer into the main channel and pure triolein into the side channel and stabilized the buffer-triolein interface 50–100 μm below the T-junction by closing the valve in the side-channel (*Figure 2A*). In this configuration, the buffer-triolein interface is not disturbed by the flow in the main channel, whereas the solutes from the main channel are free to diffuse to the oil surface. In terms of diffusion and hydrodynamic characteristics, this system is similar to microfluidic cavities (*Osterman et al., 2016*; *Vrhovec et al., 2011*).

We introduced Alexa-488-labeled Plin4 12mer into the main channel, and we followed the change in fluorescent signal inside the side channel and on the triolein interface over time using a confocal microscope (*Figure 2B* and *Videos 1–4*). As the protein solution in the main channel reached the T-junction, we could observe its diffusion into the side channel (*Video 1*). After several seconds, we detected an increase in fluorescence on the oil interface, which stabilized in ≈ 3 min at a level three-fold higher than the fluorescence of the solution (*Figure 2C*). We then replaced the protein solution in the main channel with buffer to promote protein dissociation (*Video 2*). However, the fluorescence at the interface remained constant, indicating a stable interaction between Plin4 12mer and oil. No enrichment of fluorescence on the oil interface was observed when we introduced buffer containing Alexa488 conjugated to free cysteine instead of Plin4 12mer (lower row in *Figure 2B,C* and *Videos 3* and *4*). These experiments confirm that Plin4 AH forms a very stable protein layer at the oil/water interface. Fluorescent protein also adsorbed to the glass surface and intercalated into the glass-oil interface. The chemically heterogeneous glass surface induced a marked hysteresis in oil-glass contact angle (*Joanny and de Gennes, 1984*; *Figure 2—figure supplement 1*), which allowed us to apply the Laplace law and verify that the protein adsorption lowered the surface tension of the oil interface (*Figure 2—figure supplement 1*).

We compared the interaction Plin4 12mer with oil to that of the less hydrophobic Plin4(4T > S) mutant using the microfluidics system. The mutant assembled on the oil surface with no measurable difference in the kinetics of assembly or in the factor of enrichment compared to Plin4 12mer. The difference between Plin4 12mer and Plin4(4T > S) became obvious when the two proteins were used as unlabeled proteins at a 50:1 molar excess over labeled Plin4 12mer. We observed strong fluorescent signal on the oil interface in the presence of the Plin4(4T > S) mutant, but not in the presence of Plin4 12mer, indicating that the wild-type protein out-competed with Plin4(4T > S) for oil coating (*Figure 2—figure supplement 2*).

## Comparison between the AH of Plin4 and other perilipins

So far, we focused on the interaction between the AH of Plin4 and LDs as it represents a most striking example of an LD-binding AH. We wanted to specifically compare the characteristics and LD-binding properties of Plin4 AH with the AH regions of the other human perilipins (Plin1, Plin2, Plin3), which have been shown to contribute to their LD targeting (*McManaman et al., 2003*; *Nakamura and Fujimoto, 2003*; *Bulankina et al., 2009*; *Rowe et al., 2016*). The number of 11-aa repeats that we could identify in each Plin protein ranged from five for Plin5 to about eight for Plin1/2/3, compared to the 87 repeats in Plin4. In addition, the repeats are more highly conserved in Plin4, and Plin4 AH is also striking for the absence of any deletions or insertions between the repeats (*Čopič et al., 2018*; *Figure 3A*). Comparison of the composition of the 11-aa regions showed that they were similar in character in Plin2/3/4, with low hydrophobicity due to a lack of large hydrophobic residues (*Figure 3B*). Plin1 AH is somewhat more hydrophobic and contains some aromatic residues. A more divergent character of this AH is consistent with the evolutionary divergence of Plin1 from the other perilipins (*Granneman et al., 2017*). The AH of Plin5 is shorter than in other perilipins and we did not consider it in further analysis.

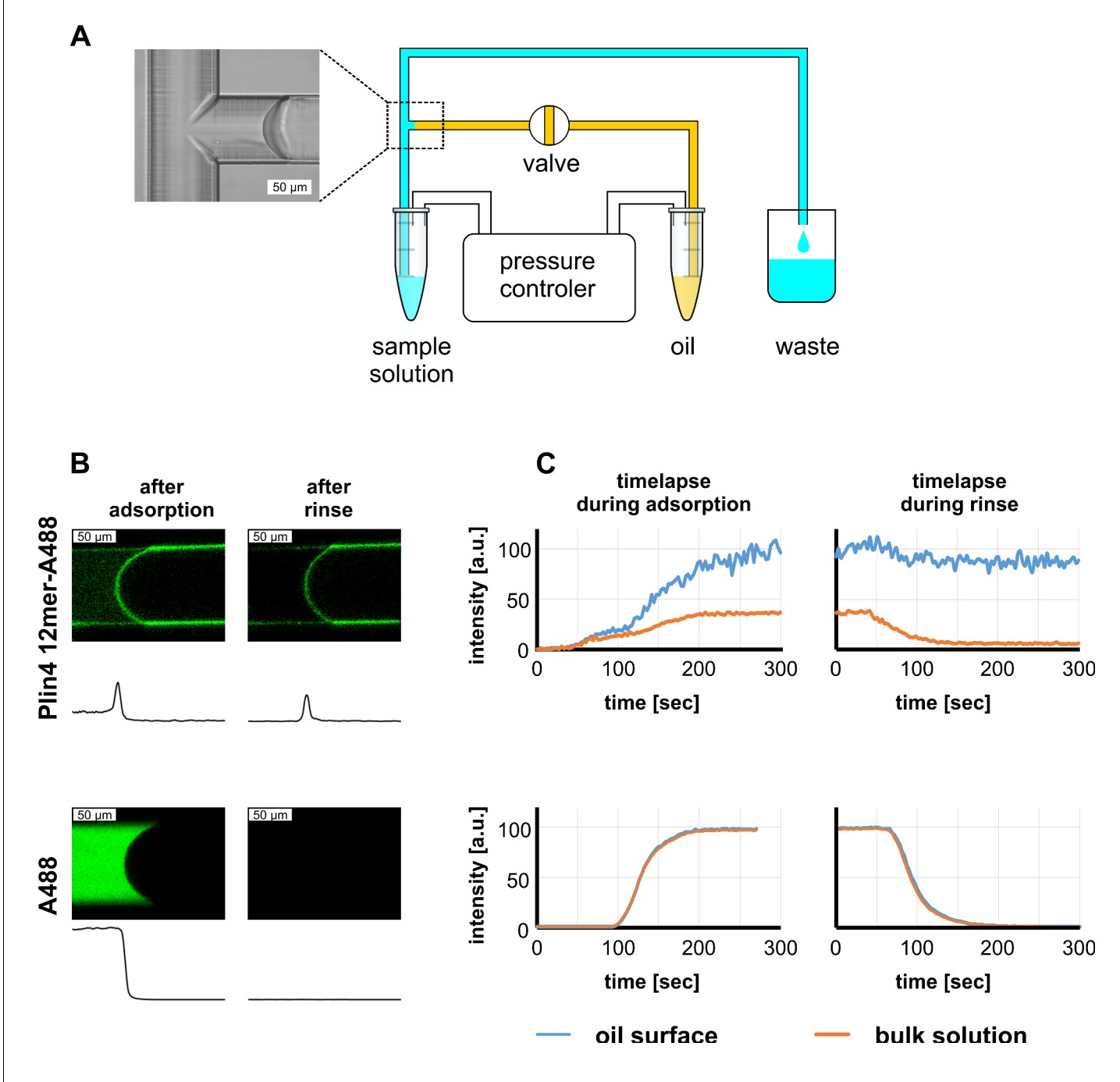

**Figure 2.** Real-time monitoring of protein–oil interaction in a microfluidic system shows irreversible adsorption of Plin4 12mer-A488 on triolein.
(**A**) Scheme of the microfluidics experimental set-up. (**B**) Top row: confocal images of the triolein-buffer interface as formed in the microfluidic system after adsorption of Alexa488-labeled Plin4 12mer on the triolein surface and after rinsing with buffer. Bottom row: control experiment with the free fluorophore Alexa488. The intensity profile along the channel center is shown below each confocal image. The protein adsorbs irreversibly at the oil surface, whereas Alexa488 conjugated to free cysteine (A488) does not. See also *Videos 1–4*. A representative of three independent experiments is shown. (**C**) Time course of the signal of Alexa488-labeled Plin4 12mer or of free Alexa-488 in the side channel as quantified from the experiment shown in B. A representative of three independent experiments is shown.

The online version of this article includes the following figure supplement(s) for figure 2:

**Figure supplement 1.** Microfluidic experiments demonstrate that Plin4 12mer lowers the surface tension of oil.

**Figure supplement 2.** Plin4 mutant 4T > S-A488 also adsorbs to the oil surface, but is outcompeted by Plin4 12mer-A488.

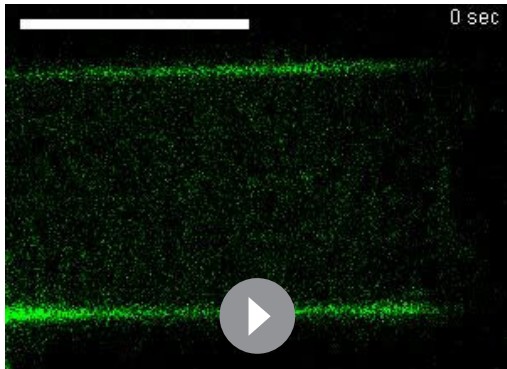

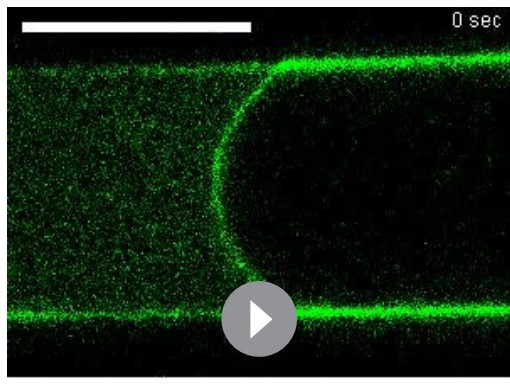

**Video 1.** Adsorption of Plin4 onto the triolein interface. A confocal time-lapse recording of diffusion of Alexa-488-labeled Plin4 12mer into the side microfluidic channel and its adsorption onto the triolein interface. The relative fluorescence intensity profile along the channel center is shown at the bottom. Scale bar: 100 μm.

https://elifesciences.org/articles/61401#video1

**Video 2.** Plin4 remains bound onto the triolein interface after rinsing. A confocal time-lapse recording of rinsing the side microfluidic channel with buffer after Alexa-488-labeled Plin4 12mer adsorbed to the triolein interface (see *Video 1*). The fluorescence intensity profile along the channel center is shown at the bottom. Scale bar: 100 μm.

https://elifesciences.org/articles/61401#video2

We first expressed Plin AHs as GFP fusions in budding yeast and assessed their ability to target LDs. Budding yeast was used previously for expression of mammalian perilipins; full-length Plin1, Plin2, and Plin3, as well as their N-terminal halves, which include a region termed 'PAT domain' in addition to 11-aa repeats, targeted LDs in this system (*Jacquier et al., 2013*; *Rowe et al., 2016*; *Čopič et al., 2018*). We expressed the AHs of Plin1, Plin2, and Plin3, and fragments of different lengths from the AH region of Plin4, containing 4, 6, or 12 33-aa repeats (132, 198, and 396 aa, respectively). In the case of Plin3, we could not observe any expression of just the AH region (aa 113–205) fused to GFP, therefore we added some additional upstream sequence (aa87-205) (*Bulankina et al., 2009*). We expressed these constructs under three growth conditions that promote LD accumulation in yeast (*Gao et al., 2017*): (i) wild-type cells grown to stationary phase; (ii) stationary phase cells lacking the most abundant yeast LD protein, Pet10p/Plin1p (*pet10Δ*); (iii) *pet10Δ* cells grown in oleic-acid rich medium, which promoted the formation of large LDs (*pet10Δ* + OA). In wild-type cells, Plin1 AH, but not Plin2 AH, Plin3 AH, or Plin4 4mer, could be observed on LDs (*Figure 3C*). In contrast to Plin4 4mer, Plin4 6mer, and Plin4 12mer localized to LDs, in line with our finding that increasing the AH length improves LD targeting (*Čopič et al., 2018*). In agreement with the work of Gao et al., deletion of Pet10p/Plin1p improved LD targeting of our mammalian constructs, presumably because more LD surface was available (*Kory et al., 2015*). Targeting to LDs was further increased by the addition of oleic acid to stationary phase cells, which induced large LDs (*Figure 3C*). In addition, we observed some protein at the PM, in particular in the case of Plin4 AH, consistent with observations from human cells and tissues (*Scherer et al., 1998*; *Ruggieri et al., 2020*). Based on these results, we conclude that the 11-aa repeat regions of Plin1, Plin2, Plin3, and Plin4 are all sufficient for targeting LDs. Comparison of different growth conditions (*Figure 3D*), and the fact that all Plin AHs were expressed at similar levels (*Figure 3—figure supplement 1B*), allowed us to establish a ranking of Plin AH-LD affinities. Extrapolating to its full length, Plin4 AH has the highest affinity for LDs, followed by Plin1 AH, and finally by Plin2 AH and Plin3 AH. However, correcting for length differences reveals that per unit of AH

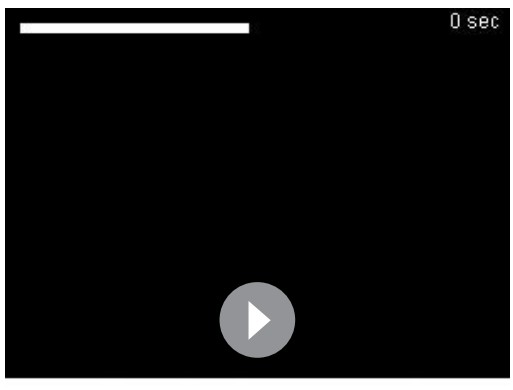

**Video 3.** Alexa-488 dye does not adsorb onto the triolein interface. A confocal time-lapse recording of Alexa-488 dye diffusion into the side microfluidic channel toward the triolein meniscus. The relative fluorescence intensity profile along the channel center is shown at the bottom. Scale bar: 100 μm.

https://elifesciences.org/articles/61401#video3

### to LDs in model cellular systems

We used FRAP to assess the stability of AH binding to yeast LDs. Plin2 AH-GFP and Plin3 AH-GFP could readily exchange between LD surface and the cytosol in cells grown for 24 hr in oleic acid medium, as reflected by a recovery half-life on the order of a few seconds (*Figure 4A*). The exchange of Plin1 AH-GFP was 2–3 times slower, consistent with results obtained in mammalian cells with N-terminal halves of Plin1, Plin2 and Plin3 (*Ajjaji et al., 2019*). In striking contrast, Plin4 12mer-GFP displayed almost no recovery on LDs over a period of more than 5 min (*Figure 4A*). We observed a similar difference between Plin1 AH and Plin4 12mer in cells grown for to early stationary phase in standard growth medium (*Figure 4B*). However, these LDs were much smaller and more mobile, leading to a large variability in the fluorescence measurement. Due to small size of these LDs, we could not perform a partial FRAP to assess the lateral mobility of Plin AH constructs on LD surface as we did for Plin4 AH on oil in vitro. However, we took advantage of the fact that both Plin4 AH and Plin1 AH also localized to the yeast PM in *pet10Δ* cells (*Figure 4C*). Bleaching a small

length, Plin1 AH has a higher affinity for LDs than AHs of Plin2, Plin3, or Plin4. This is consistent with the higher hydrophobicity of Plin1 AH compared to other perilipin AHs (*Figure 3B*); higher hydrophobicity has been shown to promote LD binding (*Čopič et al., 2018*; *Prévost et al., 2018*).

Strikingly, we noticed a difference in the size of the LDs that formed in *pet10Δ* cells grown in oleic-acid-rich medium, depending on the AH expressed (*Figure 3C,E*): LDs were significantly larger (2.5-fold difference in projected area, which would correspond to a fourfold difference in volume) when covered with Plin1, Plin2, or Plin3 AH, compared to LDs covered with Plin4 12mer (*Figure 3C,E*). LDs with Plin4 6mer were also somewhat larger than those covered with Plin4 12mer. LDs with Plin4 4mer were more variable in size and appearance, preventing the use of the same quantification protocol.

We conclude that AHs from all four perilipins (Plin1-4) can target LDs and that their affinity for LDs correlates with their length and hydrophobicity. However, the AH of Plin4 could reduce the size of LDs more strongly than the AHs of other perilipins.

## Stability of binding of perilipin AHs

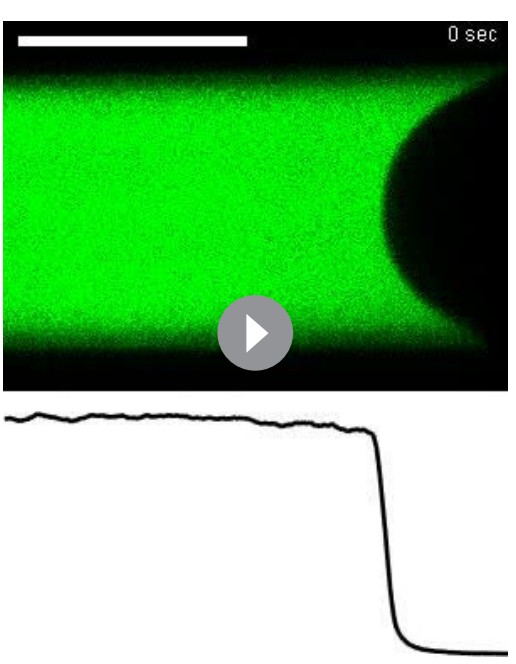

**Video 4.** Rinsing completely removes Alexa-488. A confocal time-lapse recording of rinsing Alexa-488 dye out of the side microfluidic channel. The fluorescence intensity profile along the channel center is shown at the bottom. Scale bar: 100 μm.

https://elifesciences.org/articles/61401#video4

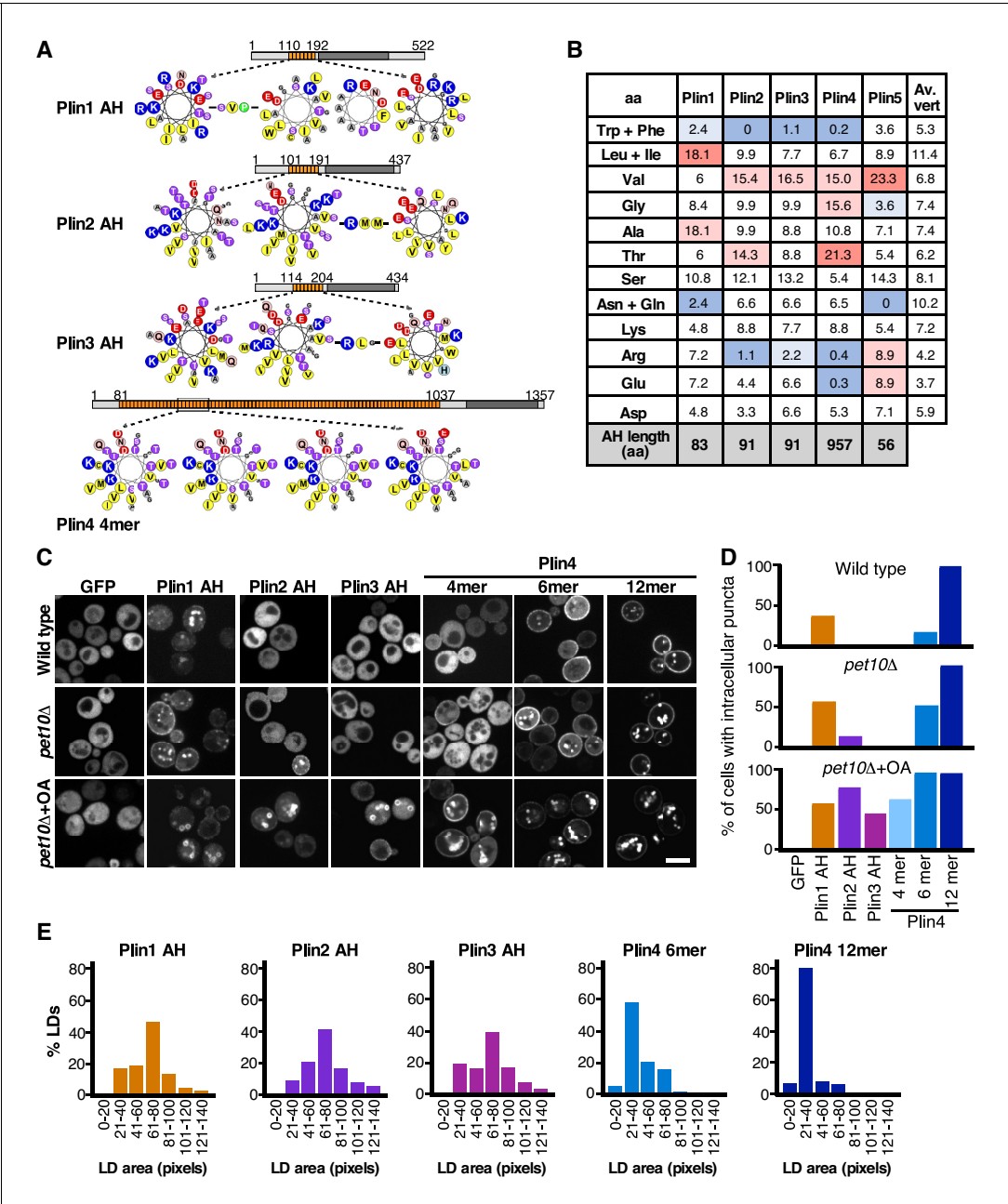

**Figure 3.** Comparison of the LD-binding properties of perilipin AHs in yeast. (**A**) Helical wheel representation of the AHs of Plin1 (aa 110–189 aa), Plin2 (aa 101–191 aa), Plin3 (aa 114–204), and Plin4 (aa 246–377, corresponding to the Plin4 4mer construct). In the case of Plin1, Plin2, and Plin3, the predicted AH regions are interspersed by short aa linkers, which are also indicated. Diagrams above the helical wheels show the full-length proteins, with AH regions shown in orange and the four-helix bundle in dark gray. (**B**) AA composition of the AH of Plin1, 2, 3, and 4 (in %) in comparison with the average aa composition of vertebrate proteins (av. vert). The blue and red backgrounds indicate lower or higher % as compared to vertebrate values, respectively. (**C**) Localization of GFP fusions with the AH region of Plin1, Plin2, Plin3, or Plin4 in *S. cerevisiae* cells. The experiment was performed with wild-type yeast cells (upper row), with *pet10Δ* cells (medium row) grown for 24 hr to stationary phase, or with *pet10Δ* cells grown to stationary phase and then transferred for 24 hr to oleic acid (OA) medium (lower row). Scale bar: 5 μm. (**D**) Bar plots of the percentage of cells showing intracellular puncta for the different proteins expressed. Sixty cells per each condition were counted in one of at least two representative experiments. (**E**) Quantification of the size distribution of fluorescent LDs (labeled with GFP-fusion proteins) in *pet10Δ* + OA cells. The plots show representative measurements from two independent experiments, where the following number of LDs was counted: Plin1 AH, 141; Plin2 AH, 143; Plin3 AH, 136; Plin4 6mer, 148; Plin4 12mer, 159. Pixel size: 0.091 μm x 0.091 μm.

The online version of this article includes the following source data and figure supplement(s) for figure 3:

**Source data 1.** Comparison of the LD-binding properties of perilipin AHs in yeast and their effect on LD size.

*Figure 3 continued on next page*

*Figure 3 continued*

**Figure supplement 1.** Plin AH fusion proteins co-localize with LDs in yeast and are expressed at similar levels.

area on the PM in exponentially growing cells showed that Plin4 12mer was highly immobile. Mobility of Plin4 AH at the PM was increased when we used shorter constructs (8mer and 6mer), in agreement with the correlation between AH length and binding affinity (*Figure 4D* and *Čopič et al., 2018*). However, even for the shortest Plin4 AH construct for which we consistently observed targeting to yeast membranes, the recovery after photobleaching was at least an order of magnitude slower than for Plin1 AH or Plin3 AH. Given that all AH constructs were expressed at similar levels (*Figure 3—figure supplement 1*), that is, the differences in the kinetics were not due to differences in protein concentration, we conclude that, in addition to its length, the particular composition of the Plin4 AH enables its stable binding to LDs.

We also tested the dynamics of Plin4 12mer-LD interaction in *Drosophila* Schneider 2 (S2) cells (*Figure 4—figure supplement 1A*). These cells were used to decipher mechanisms of LD homeostasis (*Guo et al., 2008*; *Krahmer et al., 2011*), and we previously demonstrated that expression of Plin4 12mer in S2 cells rescued the increase in size of LDs following the depletion of phosphatidylcholine (PC) (*Čopič et al., 2018*). Interestingly, we observed a large cell-to-cell variability in the FRAP recovery curves of Plin4 12mer-GFP on LDs, whereas in some cells Plin4-12mer on LDs was largely immobile (half-time of recovery >100 s), similar to our results in yeast, in other cells the recovery could be on the order of 1 s (*Figure 4—figure supplement 1A*; note that within the same cell, the signal on all LDs recovered at the same rate). PC depletion upon CCTα knocked-down had a small effect, but this was not the main driver of cell-to-cell variability (*Figure 4—figure supplement 1B*). The level of protein expression was also not very predictive of recovery rate. In contrast, we observed a correlation between the rate of FRAP recovery and the intensity of the Plin4 fluorescent signal on LDs (*Figure 4—figure supplement 1C*). This observation suggests that Plin4 AH density at the LD surface influences its dynamics, a feature reminiscent of protein coats. At low membrane coverage level, coat subunits diffuse and exchange quickly; at high membrane coverage level, their polymerization by side-side interaction prevents lateral mobility and fast turnover (*Saleem et al., 2015*; *Sorre et al., 2012*).

## Comparison of proteolipid droplets formed with Plin4 AH or Plin3 AH

To study in more detail the difference between Plin4 and other perilipin AHs binding to LDs, we used our in vitro assays to compare the behavior of purified Plin4 AH fragments with that of Plin3 AH. We chose Plin3 AH because it displayed a similar steady-state distribution in yeast as the slightly longer Plin4 4mer; however, it showed a rapid exchange between LDs and the cytosol and it did not decrease LD size in oleic acid media. Mixing purified Plin3 AH with oil resulted in a highly turbid suspension, similar to the suspensions obtained with Plin4 4mer or Plin4 12mer (*Figure 5A* and *Figure 5—figure supplement 1*). By DLS, Plin4 4mer-oil droplets behaved like Plin4 12mer oil droplets (see *Figure 1G*), displaying a particle size profile with a single peak that did not change over 14 days (*Figure 5B*, left panel). In contrast, the droplets produced by Plin3 AH were more heterogenous with larger peak sizes already 3 hr after droplet formation. Thereafter, we observed a spreading of the peaks until the samples became too complex for DLS analysis (14 days after formation *Figure 5B*, right panel). Such complexity is generally due to the presence of particles of variable sizes, suggesting that Plin3 AH-oil particles were undergoing fusion due to less stable coating by Plin3 AH. We verified using circular dichroism (CD) that this difference was not due to poor folding of the Plin3 helix (*Figure 5—figure supplement 1B*): folding of Plin3 AH in the presence of the helix-inducing reagent trifluoroethanol (TFE) was similar to what we previously showed for Plin4 AH (*Čopič et al., 2018*; see also *Figure 6—figure supplement 1*).

Centrifugation of AH-oil suspensions on sucrose gradients revealed a smaller fraction of total Plin3 AH protein associated with the oil fraction (top of the gradients) than Plin4 4mer or Plin4 12mer (*Figure 5C,D*). This could be either because less Plin3 AH was bound to the oil droplets or because Plin3 AH bound to oil less strongly and dissociated during centrifugation. To distinguish between these possibilities, we performed competition experiments in which we first formed protein-oil droplets by mixing oil with a high concentration of unlabeled purified AH constructs (Plin4

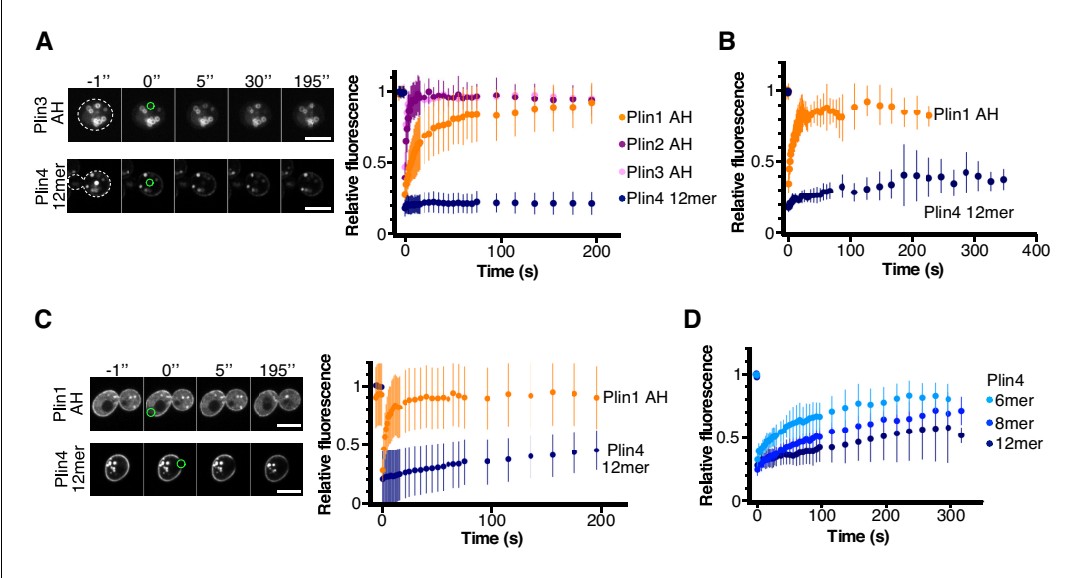

**Figure 4.** Dynamics of perilipin AH-GFP fusions on LDs or at the plasma membrane in yeast. (**A**) Dynamics of AH-GFP fusions on LDs in *pet10Δ* cells grown for 48 hr in oleic acid. Images show two representative FRAP time-courses in cells expressing Plin3 AH-GFP (top panels) or Plin4 12mer-GFP (bottom panels); time (in seconds) is indicated on top. Cells are outlined in white and the bleached areas containing one LD are outlined in green. Graph shows recovery curves for Plin1 AH-GFP (n = 29), Plin2 AH-GFP (n = 14), Plin3 AH-GFP (n = 16), or Plin4 12mer-GFP (n = 24). (**B**) Recovery curves for Plin1 AH-GFP (n = 11) or Plin4 12mer-GFP (n = 5) on LDs in *pet10Δ* cells in late exponential phase (small LDs). (**C**) FRAP of Plin1 AH (n = 14) and Plin4 12mer (n = 15) at the PM in exponentially growing *pet10Δ* cells. Images show two representative time-courses for Plin1 AH (top) and Plin4 12mer (bottom). Bleached areas are outlined in green. (**D**) FRAP of Plin4 6mer-GFP (n = 12), Plin4 8mer-GFP (n = 7) and Plin4 12mer-GFP (n = 7) at the PM in exponentially growing wild-type cells. All graphs show the mean ± SD of the fluorescence recovery curves from n FRAP measurements on different LDs or different regions of the PM, as shown in the images. Scale bar: 5 µm.

The online version of this article includes the following source data and figure supplement(s) for figure 4:

**Source data 1.** Dynamics of perilipin AH-GFP fusions on LDs or at the plasma membrane in yeast assessed by FRAP.

**Figure supplement 1.** Cell-to-cell variability in the recovery of Plin4 12mer-GFP after photobleaching of LDs in *Drosophila* S2 cells.

12mer, Plin4 4mer or Plin3 AH). Then, we gently added Alexa488-labeled Plin4 12mer at an excess mass ratio of 20:1 compared to unlabeled protein, and we monitored the fluorescence of the suspensions over time using confocal microscopy (*Figure 5E*). Consistent with our previous results, we observed no incorporation of fluorescent Plin4 12mer into the preformed Plin4 12mer-oil particles over a period of 24 hr, unless we vortexed the suspension (*Figure 5E,F*; top panel). We could observe some incorporation of fluorescent Plin4 12mer into Plin4 4mer-oil particles after 3 or 24 hr of incubation, in agreement with a more stable binding of a three-times longer AH and our cellular data (*Figure 5E,F*; middle panel). In striking contrast, when we pre-formed AH-oil particles using Plin3 AH, Plin4 12mer readily incorporated into these particles, reaching close to maximal particle fluorescence within 10 min after Plin4 12mer addition (*Figure 5E,F*; bottom panel). In agreement with the DLS data, we also observed clustering of Plin3 AH-formed oil particles, especially after 24 hr of incubation. Because Plin3 AH is only 1.5-times shorter than Plin4 4mer, yet it displays a significantly less stable binding to oil droplets in vitro and to LDs in cells, we conclude that the specific sequence of Plin4 AH is predominantly responsible for its highly stable interaction with LDs.

## The nature and distribution of aa in the polar face of Plin4 AH is critical for LD targeting

The Plin4 AH sequence displays a remarkable repetitiveness (*Figure 3A*). Positions of polar and charged residues are extremely conserved among the 33-aa repeats (*Figure 6A*). Furthermore, the sequence shows a strong preference for lysine over arginine (22-fold) and for aspartic over glutamic acid (18-fold) (*Figure 3B*). These considerations prompted us to construct mutants of Plin4 4mer in which we introduced in every 33-aa repeat modest mutations (e.g. N > Q, D > E, or K > R) that

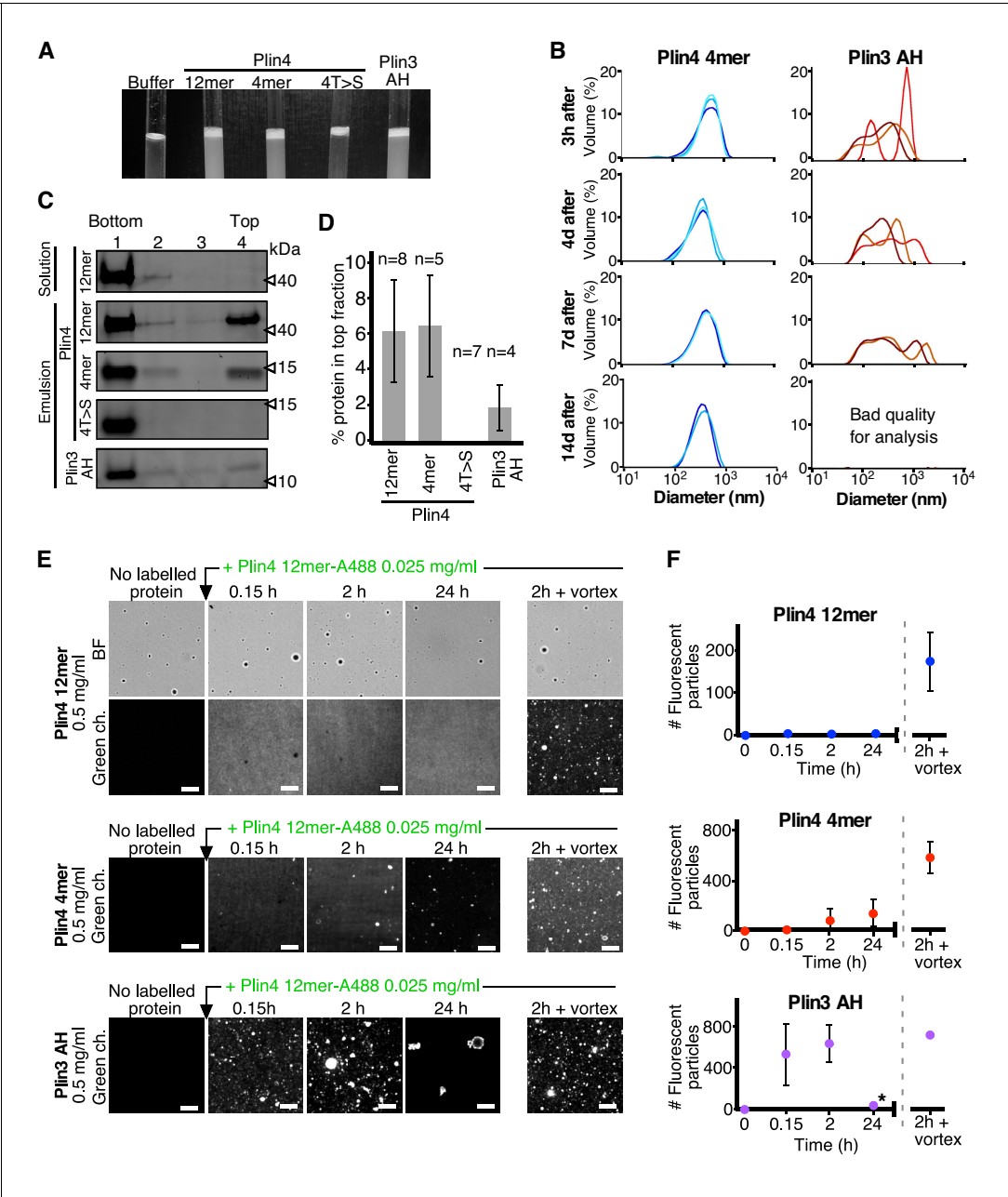

**Figure 5.** Plin3 AH interacts much less strongly with oil than Plin4 AH. (**A**) Turbidity assays with 0.5 mg/ml protein solutions of Plin4 12 mer, Plin4 4mer, Plin4 4mer mutant [4T > S], or Plin3 AH after vigorous vortexing with triolein (15 μl for 285 μl of protein solution). (**B**) Size distribution of the droplets in triolein emulsions formed with Plin4 4mer or Plin3 AH at various times after vortexing was determined by DLS. Particle size is shown by volume weighted distributions. One of three independent experiments is shown. (**C**) Protein/oil emulsions or protein solutions of the indicated variants of Plin4 or Plin3 AH were mixed with sucrose and loaded on the bottom of a sucrose step gradient. After centrifugation, four fractions were collected from the bottom and equal volumes were analyzed by protein gels with Sypro Orange staining. (**D**) Quantification (mean ± SD) of the experiment in C. The number of repeats for each experiment is indicated above the bar graphs. (**E**) Protein exchange assay in LD emulsions. Top panel: a Plin4 12mer (0.5 mg/ml)/triolein emulsion was prepared by vortexing. Thereafter, 0.025 mg/ml Alexa-488-labeled Plin4-12 mer was gently added. The emulsion was imaged at the indicated time points by light microscopy in bright field (BF) to see all particles and by fluorescence to detect coverage by Alexa-488-labeled Plin4 12 mer. Finally, the suspension was vortexed again to promote maximum incorporation of Alexa488-labeled Plin4 12mer in the emulsion. The middle and lower rows show similar experiments performed with Plin4 4mer and Plin3 AH emulsions, respectively. Scale bars: 5 μm. (**F**) Quantification (mean ± SD) of the experiments shown in E as determined from four separate fields (73 × 100 μm) in the same experiment. The graphs are representative of at least two independent experiments. Time on the x-axis is plotted using logarithmic scale. Asterisks indicates clustering of particles, which resulted a in low total number of fluorescent puncta, as seen in the image.

The online version of this article includes the following source data and figure supplement(s) for figure 5:

*Figure 5 continued on next page*

*Figure 5 continued*

**Source data 1.** Analysis of protein content and protein exchange in oil emulsions.

**Figure supplement 1.** Plin3 AH purification and CD analysis.

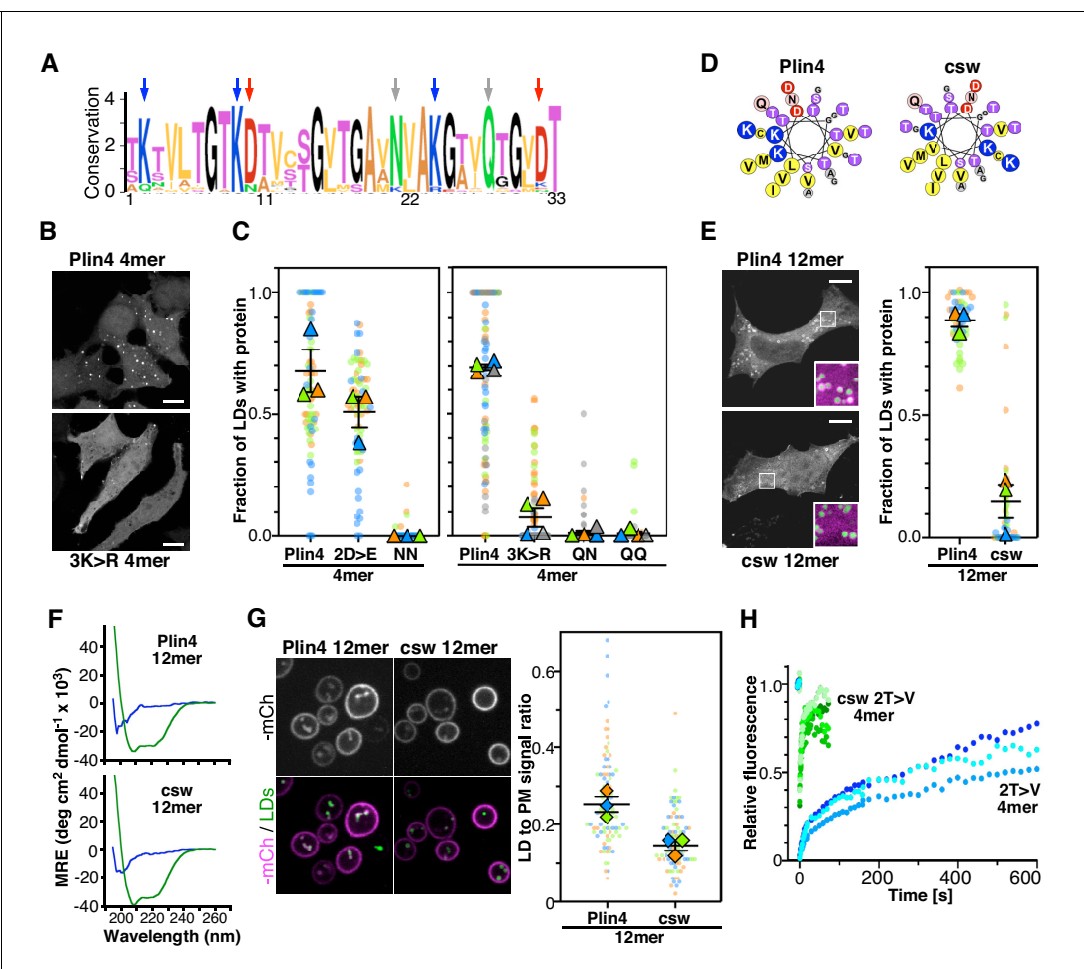

**Figure 6.** The polar face of Plin4 AH is key for specific and stable coating of LDs in cells. (**A**) Weblogo plot of the AH region of human Plin4 as determined by aligning its 29 33-mer repeats. The vertical arrows indicate the mutated aa: the NQ pair (gray), which was mutated into NN, QQ or QN; the three K (blue), which were mutated into R, and the two D (red), which were mutated into E. (**B**) Co-localization of GFP-Plin4 4mer wild-type and 3K > R (in white) with LDs (purple) in HeLa cells. (**C**) Quantification of the percentage of LDs stained with the indicated protein per cell. These 'SuperPlots' (***Lord et al., 2020***) show all data fom three to four independent experiments, each with a different color; each light dot represents one cell, whereas each triangle shows the mean from one experiment. The black bars show the mean ± SE of three to four independent experiments. (**D**) Helical wheels of Plin4 WT and csw mutant. (**E**) Localization of Plin4 12mer wild-type or csw mutant in HeLa cells. The insets show extended views with the protein in purple and LDs in green (stained with Bodipy). The super plots show the mean ± SE of the % of LDs positive for the indicated protein per HeLa cell as determined from three independent experiments. (**F**) CD spectra of Plin4 12mer (5 µM) and csw 12mer (7.5 µM) in solution (blue) or in an equal volume of buffer and TFE (green). (**G**) Light microscopy images of mCherry fusions of Plin4 12mer wild-type or csw mutant in yeast. Top: mCherry fluorescence (mCh); bottom co-localization of mCherry (purple) with LDs stained with bodipy (green). The relative fluorescence signal of mCherry fusions of Plin4 12mer wild-type or csw mutant on LDs and at the PM in *PET10-GFP* yeast strain was used to build the SuperPlots shown on the right. Data are from three independent experiments, with n ≥ 25 for each condition in each assay. (**H**) Fluorescence recovery curves of mCherry fusions of Plin4 4mer 2T > V (green points) and 2T > V csw mutants (blue points) on LDs in HeLa cells. Each curve represents FRAP of a single LD in one cell. The online version of this article includes the following source data and figure supplement(s) for figure 6:

**Source data 1.** The polar face of Plin4 AH is key for specific and stable coating of LDs in cells.

**Figure supplement 1.** Plin4 4mer mutants are not affected in their helical folding.

**Figure supplement 2.** Additional data for Plin4 charge-swap mutant.

should not modify the folding and overall physical chemistry of the helix, including its charge and hydrophobic moment.

We first focused on the two conserved amide residues: an asparagine present in 25 out of 29 repeats of human Plin4, and a glutamine seven residues apart and conserved in all repeats (*Figure 6A*). The N[x]$_6$Q sequence was replaced by N[x]$_6$N (NN), Q[x]$_6$Q (QQ) or Q[x]$_6$N (QN). Strikingly, these three mutations almost eliminated the targeting of Plin4 4mer to LDs in HeLa cells (*Figure 6C*). We purified the NN 4mer mutant to verify by CD that this mutation did not prevent helical folding. Like Plin4 4mer and Plin4(4T > S), this mutant was unfolded in solution but displayed a strong helical signal in the presence of TFE (*Figure 6—figure supplement 1*). Next, we considered the charged residues. Replacing all aspartates with glutamates (2D > E) led to a small decrease in AH targeting to LDs in HeLa cells, whereas replacing the lysine residues with arginine (3K > R) almost abolished AH targeting to LDs (*Figure 6B,C*). These results suggested that a precise interaction between charged and/or polar residues could be important for LD binding.

We also noted an unusual distribution of charged residues throughout the Plin4 AH sequence, with positive ones always lying on one side of the helix close to the apolar/polar interface (*Figure 6D*). To test whether charged residues in Plin4 AH could mediate interhelical interaction at the LD surface, we prepared a mutant of Plin4 12mer in which we reorganized the distribution of charges in the polar face of all 33-mer repeats without changing the overall composition (*Figure 6D*). This more symmetric Plin4 12mer AH mutant, termed charge-swap (csw), was similar to a 4mer mutant that we tested previously and which did not localize to LDs when expressed in HeLa cells (*Čopič et al., 2018*). We observed some localization of the longer csw 12mer-GFP mutant to LDs in HeLa cells, which was significantly reduced compared to Plin4 12mer-GFP (*Figure 6E*). We verified by CD spectroscopy using purified csw 12mer that the permutations of the sequence did not affect its helical folding (*Figure 6F* and *Figure 6—figure supplement 2A*). When we compared the localization of these two constructs in the yeast model, we observed a difference in their distribution between LDs and the PM, with csw 12mer showing a lower ratio of LD-to-PM signal compared to Plin4 12mer (*Figure 6G* and *Figure 6—figure supplement 2B*). This preference for the PM is consistent with the distribution of positive charges in the csw 12mer helix, which is optimal for mediating electrostatic interactions with the negative surface of the PM. We also noted a small shift toward a larger size LDs surrounded by csw 12mer, compared to Plin4 12mer (*Figure 6—figure supplement 2C*).

A previous analysis of the binding properties of disparate AHs to LD suggested the importance of large hydrophobic residues for LD binding (*Prévost et al., 2018*). Although the aa composition of perilipin AHs does not fit with this model, we tested whether increasing the hydrophobicity of Plin4 AH could by itself be sufficient for a highly stable LD binding, without the contribution of the polar face. For this, we compared the dynamic behavior of two previously characterized Plin4 4mer mutants in HeLa cells, a more hydrophobic 2T > V (containing in total eight substitutions in the four repeats), and 2T > V csw, which in addition has a reorganized polar face (*Čopič et al., 2018*). Strikingly, whereas 2T > V exhibited slow recovery curves by FRAP in HeLa cells, the recovery was ~60 times faster for the 2T > V csw mutant (*Figure 6H*). This change in dynamics by two orders of magnitude suggests that if hydrophobicity and electrostatics can roughly compensate for each other to promote effective targeting of AHs to LDs, only a precise distribution of charged residues can lead to the exceptional stability that is observed for Plin4 AH at the LD surface.

Finally, we tested the interaction of purified csw 12mer with oil. Like Plin4 AH, csw 12mer could produce oil droplets in our vortexing assay (*Figure 7A*). However, the droplets formed with csw 12mer were significantly larger by DLS (*Figure 7B*). When we added Alexa488-labeled Plin4 12mer to preformed csw-12mer-oil droplets, we observed a significant incorporation of Alexa-488 fluorescence into the droplets (*Figure 7C,D*), in contrast to the lack of exchange observed between nonfluorescent and fluorescent Plin4 12mer (*Figure 5E,F*). Therefore, mutations in the polar side of Plin4 AH that reorganized its charged residues led to a large decrease in the stability of Plin4 AH interaction with lipid particles or LDs and a concomitant increase in particle size, in agreement with our cellular data.

## Model of Plin4 helix on lipid surface

Based on differences in the behavior of Plin4 and other perilipin AHs and the results obtained with Plin4 AH mutants, we hypothesize that the particular aa distribution in the polar face of the Plin4 AH

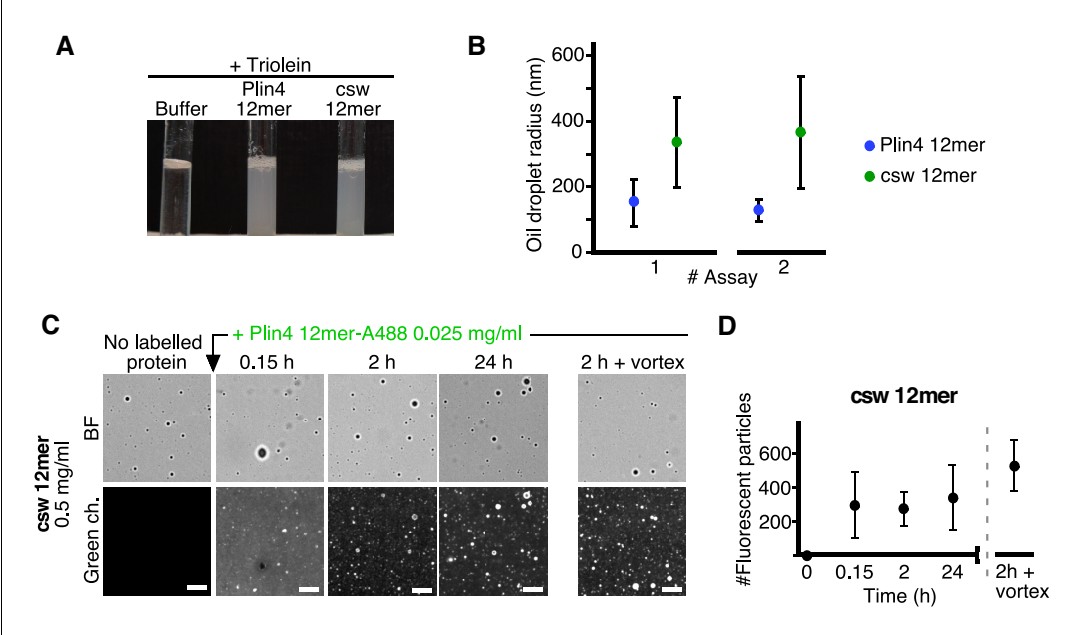

**Figure 7.** The distribution of charged residues in the polar face of Plin4 AH is key for stable coating of triolein in vitro. (**A**) Fifteen µl triolein was added to 285 µl of HK buffer containing Plin4 12mer wild-type or the csw mutant (0.5 mg/ml each). After vigorous vortexing, the samples were photographed. (**B**) DLS measurements of the hydrodynamic radius of particles in emulsions shown in (**A**), from two independent experiments (note that data in assay #1 for Plin4 12mer is the same as in *Figure 1G* at 3 hr). Dots represent peak maxima and vertical bars represent polydispersity from one representative measurement. (**C**) A triolein emulsion was prepared with Plin4 12mer or csw 12mer mutant (0.5 mg/ml). At the indicated time, fluorescent Plin4 12mer-Alexa488 (0.025 mg/ml) was gently added. The emulsions were imaged in the bright field mode (BF) and by fluorescence to detect the incorporation of Plin4 12mer-Alexa488 into the proteolipid particles. Finally, the suspension was vortexed again to promote maximum incorporation of Plin4 12mer-Alexa488 in the emulsion. Scale bars: 5 µm. (**D**) Quantification (mean ± SD) of the experiment shown in B, as determined from four separate fields (73 × 100 µm) in the same experiment. The graphs are representative of at least two independent experiments. Time on the x-axis is plotted using logarithmic scale.

The online version of this article includes the following source data for figure 7:

**Source data 1.** Redistribution of charged residues of Plin4 AH affects particle size in oil emulsions and dynamics of protein-oil interaction.

enables formation of a highly stable helical lattice on a lipid surface. To test this model, we first analyzed the structure of Plin4 AH on the surface of protein-oil particles formed in vitro. For this aim, we purified Plin4 12 mer-oil particles by centrifugation through a sucrose gradient before subjecting them to analysis by CD spectroscopy (*Figure 8A*). Despite high noise due to the turbidity of the sample, the Plin4 12 mer-oil suspensions yielded spectra typical of α-helices, with a signature peak at 222 nm. Moreover, the ellipticity signal, which depends on protein concentration, compared well with that of pure Plin4 12 mer solution with TFE within the estimated concentration range (*Figure 8A*). These experiments indicate that on the surface of oil, Plin4 AH folds into a helix, as also observed with the helix-inducing agent TFE or with liposomes containing diphytanoyl phospholipids (*Čopič et al., 2018*).

Our model predicts that the AHs should be densely and uniformly packed to enable short-range lateral interactions between specific polar/charged residues. To assess the density of Plin4 helices on the surface of oil particles, we compared the surface fluorescence of Plin4-oil particles, which was very uniform (*Figure 8B*), with that of Plin4 on bead-supported bilayers (*Figure 8C,D*). We used bead-supported bilayers (*Pucadyil and Schmid, 2010*) containing diphytanoyl phospholipids, to which Plin4 AH readily binds (*Čopič et al., 2018*). These beads are highly uniform in size, can be easily imaged and their sedimentation facilitated bulk measurements of the partitioning of Plin4, allowing us to standardize Plin4 12mer-A488 fluorescence per protein surface density. When the beads were added to a solution of 20–100 nM fluorescent Plin4 12mer at a lipid to protein molar ratio of 200–1000, they acquired within minutes a largely uniform fluorescent signal on their surface, which was stable over time (*Figure 8C*). By measuring total fluorescence in solution before and after

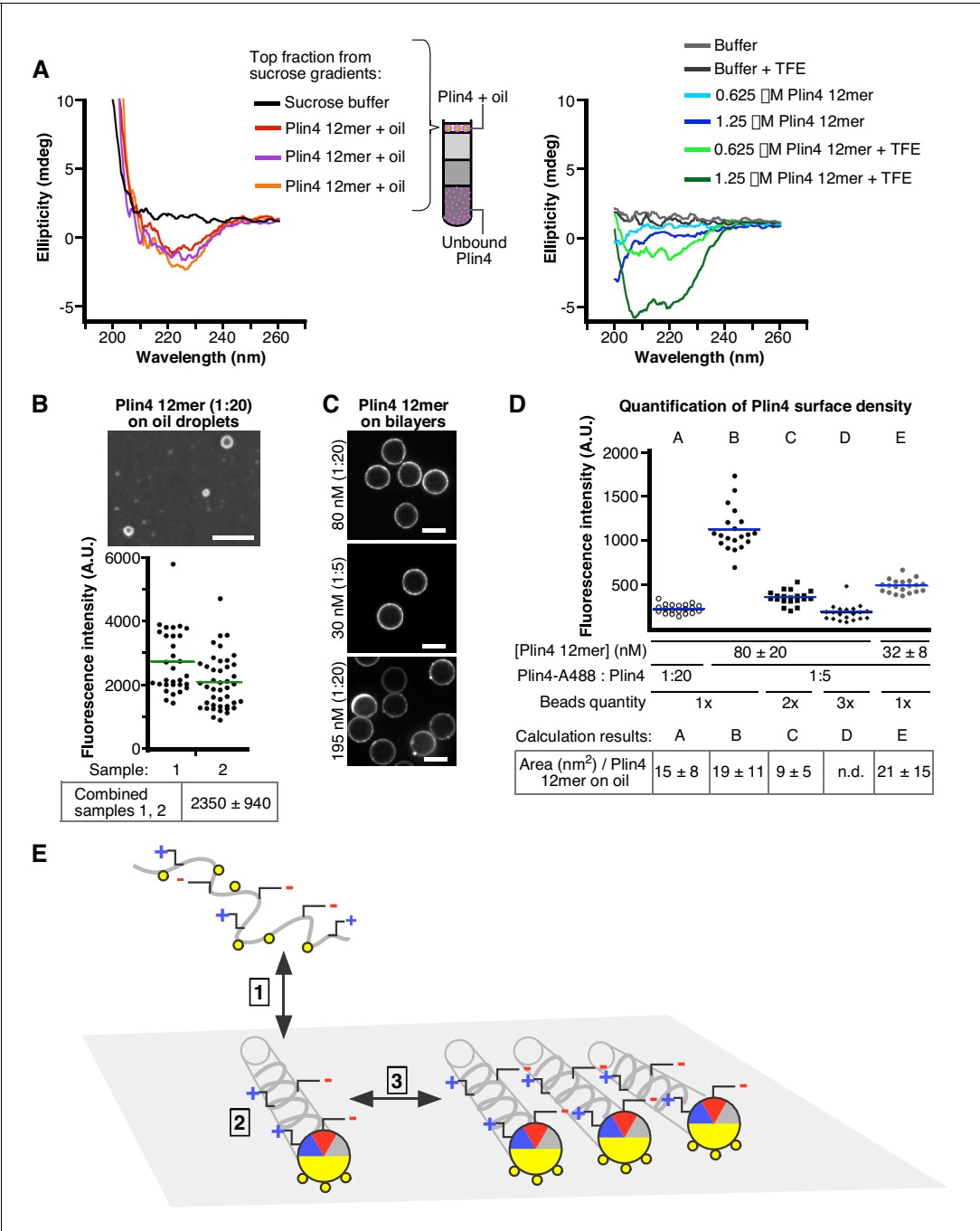

**Figure 8.** Plin4 adopts a helical conformation at the surface of oil and forms a dense monolayer. (**A**) CD analysis of three independent samples of Plin4 12mer-oil emulsion purified by sucrose step gradients (left panel), compared with CD spectra Plin4 12mer in a similar concentration range in Tris buffer (blue lines) or in buffer with 50% TFE (green lines) (**B–D**). Measurement of Plin4 12mer density at the oil surface. (**B**) A mixture of Alexa488-labeled and unlabeled Plin4 12mer (1:20) was vortexed with oil and mean droplet surface fluorescence from two independent experiments was quantified. Each data point represents one droplet. (**C**) Fluorescent Plin4 12mer mixtures at the indicated ratio and concentration were incubated with bead-supported diphytanoyl bilayers. Scale bar: 5 μm. (**D**) Graph shows mean surface bead fluorescence from five experiments (**A** to **F**) with varying Plin4 12mer concentration, Alexa488- to unlabeled Plin4 ratio and number of beads ($1x = 1.7 \times 10^6 \pm 20\%$ in 30 μl), as indicated below the graph. Each data point represents one bead; lines indicate mean values for each experiment. Bottom row: Surface area of Plin4 12mer on oil was calculated in each experiment by dividing surface fluorescence on beads per Plin4 surface density with surface fluorescence on oil (panel B); see *Supplementary file 2* for details. (**E**) Model of Plin4 AH binding to oil. 1: Plin4 AH is unfolded in solution and folds in contact with a hydrophobic surface. 2: Plin4 AH binds as a monomeric helix to the lipid surface, using its weakly-hydrophobic face. The interaction is stabilized by the length of the AH. Steps 1–2 are common for all perilipin AHs. 3: Plin4 AHs interact laterally via charged residues, forming a dense and highly stable coat-like lattice.

*Figure 8 continued on next page*

*Figure 8 continued*

The online version of this article includes the following source data for figure 8:

**Source data 1.** Estimate of Plin4 12mer density on oil surface.

incubation with beads, we determined the fraction of total Plin4 bound to beads (*Supplementary file 2*). The amount of fluorescent signal on the beads increased roughly linearly with Plin4 12mer concentration or with the concentration of fluorescent protein when we varied the ratio of fluorescent to non-fluorescent Plin4 12mer, and decreased roughly linearly with increasing number of beads (*Figure 8D*). We used the measurements of average surface fluorescence on beads at five different Plin4-to-bead ratios to calculate the fluorescence per surface Plin4 concentration. Comparing the surface fluorescence on bead-supported bilayers with surface fluorescence of Plin4-oil particles, we estimated the density of Plin4-12mer on oil at around 0.07 molecule/nm$^2$, that is, one molecule occupying an area of around 15 nm$^2$. This is close to the predicted size of a 3–11 helix 396 aa in length with the structure like the AH of α-synuclein (*Jao et al., 2008*), which would occupy an area of 30 nm$^2$ (0.5 nm x 0.15 nm x 396 aa); therefore, tight packing of such molecules falls within the range of our measurement.

We also note that the fluorescence of Plin4 12mer on oil was about 10-fold higher than the fluorescence that we could achieve on bead-supported bilayers at the same fluorescent protein dilution (1:20; see *Figure 8B–D*). At higher protein concentrations, the Plin4 AH fluorescence on beads became highly uneven, suggesting formation of domains and distortion of the bilayer (*Figure 8C*, bottom image). This was reminiscent of the behavior of α-synuclein on liposomes of a similar composition (*Garten et al., 2015*). We therefore conclude that Plin4 AH can achieve a higher density of packing on the surface of neutral lipids than in the presence of phospholipids.

We can summarize our data with a model presented in *Figure 8E*. Plin4 AH, as well as other perilipin AHs, is unfolded in solution/the cytosol. In contact with a hydrophobic lipid surface, they fold into a helix, whose apolar side engages in hydrophobic interactions with the lipid surface. Due to the low hydrophobicity of these AHs, these interactions are weak, and yet sufficient for targeting LDs due to the high length of perilipin AHs. In the case of Plin4, the helices have a high propensity to interact via polar/charged side-chains, thus forming a highly stable coat on the lipid surface.

## Discussion

Perilipins are among the most abundant proteins in the LD proteomes (*Brasaemle et al., 2004*; *Sztalryd and Brasaemle, 2017*). Whereas their importance for LD metabolism has been known for a long time, notably for the recruitment of lipases and their inhibitors, this does not explain their abundance on the LD surface. The role of Plin4, by far the largest of perilipins, has been particularly puzzling. Plin4 is a mammalian-specific protein and is highly expressed in adipocytes (*Wolins et al., 2003*). Its deletion in a mouse model has so far not revealed any strong phenotypes (*Chen et al., 2013*). However, the striking features of Plin4 AH in terms of its length, repetitiveness and particular aa composition suggest a strong selection for a specific function. Our experiments show that the interaction of Plin4 AH with LDs in vitro and in cellular model systems is remarkably stable. The slow lateral diffusion and the very slow dissociation of Plin4 AH molecules at the LD surface as assessed by FRAP, by microfluidics, or by exchange assays, are reminiscent of the behavior of vesicular coat components that polymerize on a membrane surface via lateral interactions (*Saleem et al., 2015*; *Sorre et al., 2012*).

We previously showed that the extreme length and the low hydrophobicity of Plin4 AH contributed to the specificity of its LD targeting (*Čopič et al., 2018*). Large residues such as F or W are rare or absent in the Plin4 AH sequences, whereas three small hydrophobic residues, V, T, and A, are extremely abundant (*Figure 3B* and *Figure 8*). Mutations that slightly increased hydrophobicity (T > V) made Plin4 promiscuous for other organelles besides LDs, whereas mutations that slightly decreased hydrophobicity (T > S) made Plin4 cytosolic. We now show that the Plin4(4T > S) mutant is also unable to emulsify oil in vitro although it binds to an exposed oil surface in microfluidics experiments. Thus, the hydrophobicity of Plin4 AH appears at the threshold of promoting LD binding. Our experiments in the yeast model (*Figure 3C–E*) show that this is also the case for other

perilipin AHs, consistent with their overall similar chemistries. A slight exception is the AH of Plin1, which contains some aromatic residues and partitions to LDs better than other AHs (*Rowe et al., 2016*; *Ajjaji et al., 2019*). Overall, the hydrophobicity of perilipin AHs appears at best modest and generally extremely low. As such, these proteins challenge a recent model for AH-LD interaction in which the main driving force is the intercalation of bulky hydrophobic residues within lipid packing defects at the LD surface (*Prévost et al., 2018*). This model derives from binding experiments and cellular observations performed with AHs that are much shorter than those of perilipins and bind not only to LDs but also other organelles (e.g. ALPS motif, CCTα, Arf1). As such, this model does not account for the particular chemistry of Plin AHs, which are the most abundant and specific AHs of LDs.

Surprisingly, the polar face of Plin4 AH makes a very large contribution to LD targeting. The sequence conservation in this face, including a strong preference for K over R and D over E, suggests that polar and charged residues play an important role. All mutations that we tested, including N > Q, Q > N, K > R, and D > E, decreased Plin4 AH LD targeting, with the first three mutations almost completely preventing LD binding. Furthermore, merely changing the distribution of these residues in the polar face led to a reduction in LD targeting and in stability of binding to triolein droplets in vitro. We thus propose that binding of Plin4 AH is controlled by the numerous electrostatic/hydrogen interactions that its polar side chains can engage in along its gigantic length. These features speak in favor of a 'coat' model: numerous Plin4 molecules held together by side-side interactions would form a network at the LD surface. A prediction of this model is that Plin4 should be immobilized at the LD surface and, thereby, should exhibit very limited dynamics. This was confirmed by our experiments, which revealed a drastic difference between Plin4 AH and other perilipin AHs in cells and in vitro. Furthermore, we showed that Plin4 AHs were densely packed on the surface of oil particles, in agreement with our model. Both on LDs in cells and with oil particles formed in vitro, Plin4 AH imposed a smaller size on these structures compared to other perilipin AHs or the csw mutant; therefore, stable binding of Plin4 AH correlated with smaller LD size. Interestingly, endogenous Plin4 was observed localizing to small LDs in cultured adipocytes, in contrast to Plin1 (*Wolins et al., 2003*).

The coat model of Plin4 is reminiscent of the interactions that apolipoproteins engage in to form secreted lipoprotein particles (*Phillips, 2013*). ApoA1, for which most structural information is available, forms a ring around the lipid core in low-density lipoprotein particles, stabilizing itself via interactions between charged residues from two adjacent molecules (*Bibow et al., 2017*; *Pourmousa et al., 2018*; *Melchior et al., 2017*). Twenty-six intermolecular salt-bridges connect two antiparallel rings of ApoA1, which is about 200 aa long. The abundance of positively (3) and negatively (2) charged residues in each 33-aa repeat of Plin4 is compatible with the formation of a similar large network of intra or intermolecular interactions. Interestingly, this model does not impose a strict geometry for the protein network. The large number of glycine residues in Plin4 AH (5 G per 33-aa repeat) could enable the formation of various turns, resulting in a spaghetti-like layer rather than a geometrically well-defined assembly.

The observation that perilipins decorate different LDs in the same cell type supports our model of a higher order perilipin organization (*Hsieh et al., 2012*; *Wolins et al., 2005*). The network of electrostatic interactions between perilipin AHs should be strongly dependent on their exact sequences, making the formation of hybrid coats with different perilipins less likely than homogenous perilipin coats. However, what drives the sequential coating of LDs by different perilipins remains mysterious. In addition to their repetitive AH regions, other segments of Plin1, Plin2, and Plin3 have been implicated in binding to LDs, in particular the C-terminal four-helix bundle (*Subramanian et al., 2004*; *Mirheydari et al., 2016*; *Ajjaji et al., 2019*), but this has not been the case for Plin4 (*Čopič et al., 2018*). In contrast to the behavior of their AHs, the association of full-length Plin1 and Plin2 with LDs can be very stable (*Targett-Adams et al., 2003*; *Soni et al., 2009*; *Pataki et al., 2018*; *Ajjaji et al., 2019*), and the COPI machinery has been implicated in the recruitment of Plin2 to LDs by an unknown mechanism (*Nakamura et al., 2004*; *Soni et al., 2009*).

Whereas vesicular coats uniformly cover the surface of vesicles, this is unlikely to be the case for perilipin coats. Non-uniform distribution of Plin1 on LD surface has been observed in cultured adipocytes (*Blanchette-Mackie et al., 1995*; *Hansen et al., 2017*); this can be explained by a coating model, where patches of laterally-interacting perilipin coat might coexist with LD regions decorated by other proteins. More generally, deciphering the molecular arrangements of Plin4 molecules or

other perilipins on LDs is a considerable challenge for the future. In the case of apolipoproteins, a consensual model for their organization is just starting to emerge despite decades of intense investigations on these proteins.

The sequence of Plin4 is particularly striking in light of a recent study that identified an increased number of homologous repeats (10 additional repeats) in the Plin4 AH encoding region of individuals from a single family with a rare muscular degeneration (*Ruggieri et al., 2020*). The study proposes that a higher number of repeats in Plin4 leads to protein aggregation in muscle cells, suggesting that the sequence of Plin4 represents a risk for the organism. It is then even more surprising that a protein with such a long and repetitive AH has arisen during the mammalian evolution. While our results suggest that such a configuration represents a means of guarding the stability of LDs, the precise advantages for mammalian cellular metabolic pathways remain to be discovered.

# Materials and methods

**Key resources table**

| Reagent type (species) or resource | Designation | Source or reference | Identifiers | Additional information |
|---|---|---|---|---|
| Gene (*Homo sapiens*) | PLIN1 | *Jacquier et al., 2013* | PLIN1_HUMAN AAH31084.1 | *Supplementary file 1* |
| Gene (*Homo sapiens*) | PLIN2 | *Jacquier et al., 2013* | PLIN2_HUMAN AAH05127.1 | *Supplementary file 1* |
| Gene (*Homo sapiens*) | PLIN3 | *Jacquier et al., 2013* | PLIN3_HUMAN AAC39751.1 | *Supplementary file 1* |
| Strain, strain background (*Escherichia coli*) | BL21(DE3) | ThermoFisher | C600003 | Chemically competent cells |
| Strain, strain background (*Saccharomyces cerevisiae*) | BY4742 | *Euroscarf* | MATα his3Δ1 leu2Δ0 lys2Δ0 ura3Δ0 | |
| Strain, strain background (*S. cerevisiae*) | *pet10Δ* | *Euroscarf* | MATα his3Δ1 leu2Δ0 lys2Δ0 ura3Δ0 pet10Δ::KANMX4 | |
| Strain, strain background (*S. cerevisiae*) | Pet10-GFP | *Huh et al., 2003* | MATα his3Δ1 leu2Δ0 lys2Δ0 ura3Δ0 PET10-GFP::HisMX | |
| Cell line (*D. melanogaster*) | S2 | ThermoFisher | R69007 | |
| Cell line (*Homo sapiens*) | HeLa | ATCC | CCL-2 | |
| Antibody | Anti-GFP (rabbit polyclonal) | Thermo Fisher Scientific | A11122 | (1:5000) |
| Antibody | Anti-rabbit (goat polyclonal, HRP conjugate) | Sigma-Aldrich | A6154 | (1:5000) |
| Antibody | Anti-Vps10 (mouse monoclonal) | Molecular probes | A-21274 | (1:100) |
| Antibody | Anti-mouse (donkey, HRP conjugate) | GE Healthcare | NA934V | (1:10000) |
| Peptide, recombinant protein | Plin4 12mer | *Čopič et al., 2018* | Human Plin4 (aa510-905) | *Supplementary file 2* |
| Peptide, recombinant protein | Plin4 4mer | *Čopič et al., 2018* | Human Plin4 (aa246-377) | *Supplementary file 2* |

*Continued on next page*

*Continued*

| Reagent type (species) or resource | Designation | Source or reference | Identifiers | Additional information |
|---|---|---|---|---|
| Peptide, recombinant protein | Plin4(4T > S) (4mer) | This study | | *Supplementary file 2* |
| Peptide, recombinant protein | Plin4(NN) (4mer) | This study | | *Supplementary file 2* |
| Peptide, recombinant protein | Plin4 csw (12mer) | This study | | *Supplementary file 2* |
| Peptide, recombinant protein | Plin3 AH | This study | Human Plin3 (aa113 – 205) | *Supplementary file 2* |
| Chemical compound, drug | SyproOrange | ThermoFisher | S6651 | |
| Chemical compound, drug | Alexa488 C5 maleimide | ThermoFisher | A10254 | |
| Chemical compound, drug | Alexa568 C5 maleimide | ThermoFisher | A20341 | |
| Chemical compound, drug | Bodipy 493/503 | ThermoFisher | 11540326 | |
| Chemical compound, drug | Diphytanoyl-phosphatidyl serine | Avanti Lipids | AVA-850408C-25Mg | |
| Chemical compound, drug | Diphytanoyl-phosphatidyl choline | Avanti Lipids | AVA-850356C-200Mg | |
| Other | Glass microfluidic chip with a T-junction | Dolomite | part # 3000086 | |
| Other | Glass microfluidic chip with a T-junction | Dolomite | part # 3000024 | |
| Other | Silica Microspheres, 5.00 µm, SS05N | Bang laboratories | SS05003-0.5 | |

## Sequence analysis

The 11-aa repeats of perilipins were identified using HHrepID tool from the MPI Bioinformatics Toolkit server (*Biegert and Söding, 2008*; *Zimmermann et al., 2018*). The amphipathic character of these sequences was analysed using HeliQuest (*Gautier et al., 2008*). Helical wheels were plotted as complete 3–11 helices; the presentation of helices was chosen such as to maximise their hydrophobic moment, as calculated by Heliquest, and inclusion of identified 11-aa repeats, excluding helix-breaking proline (*Pace and Scholtz, 1998*) from the middle of the helices. The amino acid conservation of the 33-aa repeats of Plin4 was represented using Weblogo (*Crooks et al., 2004*).

## Plasmid DNA construction

All plasmids used in this study are listed in *Table 1*. DNAs encoding AHs of human Plin1, Plin2 and Plin3 were PCR-amplified from the corresponding cDNAs that had been cloned into pGREG576 plasmids (gift from R. Schneiter, U. of Fribourg) (*Jacquier et al., 2013*). DNA for Plin4 6mer and Plin4 8mer was amplified from plasmid pCLG26, and DNA for Plin4 4mer mutant 4T > S was amplified from plasmids pSB49 (*Čopič et al., 2018*). Plin4 4mer mutants (2D > E, 3K > 3, NN, QN, and QQ), and Plin4 12mer mutant csw 12mer were constructed using synthetic double-stranded DNA

**Table 1.** Plasmids used in this study.

| Name | Insert | Region (aa) * | Vector | Host [†] | Source |
|---|---|---|---|---|---|
| pCLG03 | Plin4 4mer | hPlin4(246-377) | pET21b | E. coli | *Čopič et al., 2018* |
| pKE23 | Plin4 12mer | hPlin4(510-905) | pET21b | E. coli | *Čopič et al., 2018* |
| pSB49 | 4T > S (4mer) | 4x[246–278 M5t] [‡] | pmCherry-N1 | Mamm | *Čopič et al., 2018* |
| pMGA9 | 4T > S (4mer) | 4x[246–278 M5t] [‡] | pET21b | E. coli | This study |
| pGFP-Plin1 | Human Plin1 | Full cDNA | pGREG576 (ADH1pr, GFP) | Yeast | *Jacquier et al., 2013* |
| pGFP-Plin2 | Human Plin2 | Full cDNA | pGREG576 (ADH1pr, GFP) | Yeast | *Jacquier et al., 2013* |
| pGFP-Plin3 | Human Plin3 | Full cDNA | pGREG576 (ADH1pr, GFP) | Yeast | *Jacquier et al., 2013* |
| pRHT140 | ADHpr-mcs-GFP | | pRS416 (CEN-URA3) | Yeast | S. Leon |
| pMGA4 | ADHpr-mcs-mCherry (swap of GFP in pRHT140) | | pRS416 (CEN-URA3) | Yeast | This study |
| pMGA11 | Plin1 AH-GFP | hPlin1(aa108-194) | pRHT140 | Yeast | This study |
| pMGA10 | Plin1 AH-mChe | hPlin1(aa108-194) | pMGA4 | Yeast | This study |
| pMGA6 | Plin2 AH-GFP | hPlin2(aa100-192) | pRHT140 | Yeast | This study |
| pMGA5 | Plin2 AH-mCherry | hPlin2(aa100-192) | pMGA4 | Yeast | This study |
| pMGA7 | Plin3 AH-GFP | hPlin3(aa113-205) | pRHT140 | Yeast | This study |
| pMGA29 | Plin3(87-205)-GFP | hPlin3(aa87-205) | pRHT140 | Yeast | This study |
| pMGA28 | Plin3(87-205)-mCherry | hPlin3(aa87-205) | pMGA4 | Yeast | This study |
| pMGA19 | Plin3 AH | hPlin3(aa113 – 205) | pET21b | E. coli | This study |
| pKE31 | Plin4 4mer-GFP | hPlin4(aa246-377) | pRHT140 | Yeast | *Čopič et al., 2018* |
| pKE33 | Plin4 12mer-GFP | hPlin4(aa510-905) | pRHT140 | Yeast | *Čopič et al., 2018* |
| pMGA16 | Plin4 12mer-mCherry | hPlin4(aa510-905) | pMGA4 | Yeast | This study |
| pCLG26 | Plin4 8mer | hPlin4(aa246-509) | pmCherry-N1 | Mamm | *Čopič et al., 2018* |
| pMGA31 | Plin4 6mer-GFP | hPlin4(aa246-433) | pRHT140 | Yeast | This study |
| pMGA23 | Plin4 8mer-GFP | hPlin4(aa246-509) | pRHT140 | Yeast | This study |
| pACJ22 | Plin4 4mer | hPlin4(aa246-377) | pmCherry-N1 | Mamm | *Čopič et al., 2018* |
| pSB58 | 2D > E (4mer) | 4x[246–278 M17e] [‡] | pmCherry-N1 | Mamm | This study |
| pSB60 | NN (4mer) | 4x[246–278 M18q] [‡] | pmCherry-N1 | Mamm | This study |
| pMGA32 | NN (4mer) | 4x[246–278 M18q] [‡] | pET21b | E. coli | This study |
| pSB83 | QN (4mer) | 4x[246–278 M19qn] [‡] | pmCherry-N1 | Mamm | This study |
| pSB65 | QQ (4mer) | 4x[246–278 M20q2] [‡] | pmCherry-N1 | Mamm | This study |
| pSB86 | 3K > R | 4x[246–278 M21r] [‡] | pmCherry-N1 | Mamm | This study |
| pCLG62 | Plin4 12mer | hPlin4(aa510-905) | pmCherry-N1 | Mamm | *Čopič et al., 2018* |
| pSB06 | csw 12mer | Charge swap of hPlin4(aa510-905) | pmCherry-N1 | Mamm | This study |
| pCLG36 | Plin4 2T > V 4mer | 4x[246–278 M10t2] [‡] | pmCherry-N1 | Mamm | *Čopič et al., 2018* |
| pACJ41 | Plin4 csw 2T > V 4mer | Charge swap and 2T > V of 4x[246–278 M7kt] [‡] | pmCherry-N1 | Mamm | *Čopič et al., 2018* |
| pSB41 | Plin4 12mer-GFP | hPlin4(aa510-905) | pMTWG | Dros. | *Čopič et al., 2018* |
| pMGA3 | csw 12mer | Charge swap of hPlin4 (aa510-905)-GFP | pRHT140 | Yeast | This study |
| pMGA17 | csw 12mer | Charge swap of hPlin4 (aa510-905)-mCherry | pMGA4 | Yeast | This study |
| pMGA1 | csw 12mer | Charge swap of hPlin4(aa510-905) | pET21b | E. coli | This study |

* Position of amino acids (aa) in human perilipin sequences.

[†] Mamm: mammalian cells, Dros.: *Drosophila* cells.

‡ All mutants are four repeats of the same amino acid sequence.

fragments (*Supplementary file 1*). All 4mer mutants were exact 4x repeats of a 33-aa sequence, based on the parental sequence of human Plin4 fragment aa246-278. The protein sequence for csw 12mer was designed by manually adjusting 33-aa helical wheels of the parental Plin4 12mer sequence using HeliQuest to increase the symmetry of charged residue distribution in the polar side of the helix while minimizing changes in the hydrophobic moment. DNA sequences were optimized for synthesis using the algorithm on the Eurofins website (https://www.eurofinsgenomics.eu). *Supplementary file 1* also lists all protein sequences used in this study.

For expression of proteins in *E. coli*, PCR-amplified DNA fragments were inserted into pET21b (Novagen) without adding a tag using *NheI* and *XhoI* restriction sites, which were introduced by PCR. For expression of Plin3 AH, an additional sequence 'MASC' was introduced upstream of the AH.

For expression of GFP fusion proteins in *S. cerevisiae*, PCR-amplified DNA fragments were inserted into pRS416-derived (URA3 and AmpR markers) CEN plasmid pRHT140 containing ADH1 promoter and GFP for C-terminal tagging (gift from S. Leon, IJM). For expression of mCherry fusion proteins, GFP-encoding fragment in this vector was replaced with mCherry using *KpnI* and *BamHI* restriction sites to generate plasmid pMGA4. All AH DNA fragments were cloned into these plasmids using *NheI* and *BamHI* restriction sites that were introduced by PCR. The sequence of the multiple cloning site introduces a linker peptide in the resulting fusion protein between the AH and GFP, 'PLDPPGLQEF', and linker peptide 'VKDPDIKLID' between the AH and mCherry.

Plasmids for expression of mCherry fusion proteins in HeLa cells were constructed by subcloning synthetic genes for Plin4 mutants into pmCherry-N1 (Invitrogen) using *BamHI* and *XhoI* restriction sites. All plasmids were verified by sequencing.

## Protein purification

All proteins were purified from *E. coli* without a tag. Plin4 12mer and Plin4 4mer were purified as previously described (*Čopič et al., 2018*). Plin3 AH (aa103 to 205), Plin4 4T > S, Plin4 NN, and csw 12mer were purified following a similar protocol, with some modifications in the case of Plin 3 AH, as outlined below. *E. coli* cells BL21DE3 transformed with expression plasmids were grown to O.D. ≈ 0.6 at 37°C from a liquid preculture and induced with 1 mM IPTG for 1 hr at 37°C. Cells from 0.25 l cultures were collected by centrifugation and frozen. The bacterial pellets were thawed in lysis buffer (50 mM Tris-HCl pH 7.5, 150 mM NaCl, 1 mM DTT, supplemented with 0.1 mM PMSF, and Complete protease inhibitor cocktail (Roche)). Cells were broken by sonication. The lysate was centrifuged at 100,000 × g for 30 min at 4°C in a 70.1Ti Rotor (40,000 rpm; Beckman). The supernatant in centrifuge tubes was immersed in boiling water (95°C) for 30 min. The resulting cloudy suspension was centrifuged at 100,000 × g for 15 min at 4°C to remove precipitated material. The supernatant was dialyzed against 20 mM Tris-HCl pH 7.5, 10 mM NaCl, 1 mM DTT at 4°C using Spectra/Por membranes with a cut-off of 6000 Da (Spectrum labs) and then centrifuged again at 100,000 × g for 30 min at 4°C. Plin4 4T > S and csw 12mer were then further purified in a single step by cation exchange chromatography on a 20 ml Hiprep S HP column (GE Healthcare), as described for Plin4 AHs (*Čopič et al., 2018*). Plin4 NN was dialyzed against 20 mM MES-HCl pH 5.5, 10 mM NaCl, 1 mM DTT before cation exchange chromatography and eluted with a salt gradient from 10 mM to 400 mM (three column volumes) in MES buffer. Purified Plin4 NN was then dialyzed against 20 mM Tris-HCl pH 7.5, 100 mM NaCl, 1 mM DTT using a Slide-A-Lyzer dialysis cassette (Thermo Scientific) with a cut- off of 2000 Da. In contrast to Plin4 AH, Plin3 AH has a net negative charge at neutral pH (pI = 4.65). Therefore, Plin3 AH was purified by anion exchange chromatography on a 20 ml Hiprep Q HP column (GE Healthcare). It was eluted with a salt gradient from 10 mM to 400 mM NaCl (three column volumes) in 20 mM Tris-HCl pH 7.5, 1 mM DTT at a flow rate of 2 ml/min using an Akta purifier system (GE Healthcare), eluting at approximately 100 mM NaCl. After analysis of the chromatography fractions by protein electrophoresis, the protein pools were divided in small aliquots and stored at –80°C.

## Protein electrophoresis and determination of protein concentration

Standard Glycine SDS-PAGE was used for the analysis of Plin4 12mer and csw 12mer (Mw ~40 kDa) using homemade 13% acrylamide-bisacrylamide gels. Tricine SDS-PAGE (*Schägger and von Jagow, 1987*) was used for proteins with lower molecular weight, i.e. Plin4 4mer, Plin4 4T > S, Plin4 NN, or Plin3 AH (9–15 kDa). For that we either used TruPAGE commercial gels (Sigma) and homemade Tris-MOPS buffer (60 mM Trizma, 30 mM 4-Morpholinepropanesulfonic acid (MOPS), 0.1% w/v SDS), or, for better resolution, homemade 16.5% acrylamide-polyacrylamide (29:1) gels run with tricine buffer (100 mM Tris-HCl pH 8–8.5, 0.1 M Tricine, 0.1% SDS) in the cathode and 200 mM Tris HCl pH 8.9 in the anode chamber. Gels were rinsed in 7.5% acetic acid, stained with SyprOrange (Life Technologies) and visualized with a MP imaging system (Bio-Rad) using the Alexa 488 settings. Because all perilipin AH purified construct lack aromatic residues, preventing protein quantification by UV spectroscopy or by Bradford assay, protein concentration was routinely determined by densitometry of Sypro-Orange or Coomassie Blue stained gels against a calibration curve with protein standards (Sigma) using ImageJ. Quantification by gel electrophoresis was verified by Ellman´s reaction method as previously described (*Čopič et al., 2018*).

## Protein labeling with fluorescent probes

Purified AHs were covalently labeled via cysteines using Alexa C5 maleimide probes (either 488 or 568; Thermofisher). Plin4 12mer, Plin4 4mer, and Plin4 4T > S were labeled on endogenous cysteines present in their AHs; they all contain four cysteines in total. Plin3 AH is devoid of cysteines, therefore a single cysteine was introduced at its N-terminus. To remove DTT, 1 ml of protein solution at concentration 0.7 mg/ml (18 μM of Plin4 12mer and csw 12mer, 50 μM of Plin4 4mer and Plin4 4T > S) was exchanged into labelling buffer (20 mM Tris-HCl pH 7.5, 100 mM NaCl) using size exclusion NAP10 columns (GE Healthcare). Protein-containing fractions were identified by protein electrophoresis and pooled. Protein solutions were incubated for 5 min at 4°C with Alexa C5 maleimide probes at an equimolar ratio to their total number of cysteines (1 ml reaction volume). The reactions were stopped by the addition of DTT to 10 mM final concentration and loaded on NAP10 columns to separate the labeled protein from the excess of fluorescent probe. Fractions were analysed by protein electrophoresis. Fluorescence in the gel was directly visualized on ChemiDoc MP imaging system (Bio-Rad) either with Alexa 488 or SyproRuby (for Alexa 568) settings. Fractions with labeled protein were pooled, aliquoted, and stored at −80°C. The same protocol was used for labeling of free cysteine, but without NAP10 purification steps.

We used FRAP assays on protein-oil emulsions (see below) with different ratios of labeled to unlabeled proteins to verify that the fluorescent label did not change the behavior of the protein. This was not the case for labeled Plin3 AH, thus we only used this protein in unlabeled form in our biochemical assays.

## Preparation of protein-oil emulsions

Proteins were diluted to 0.5 mg/ml in freshly degassed HKM buffer (50 mM Hepes-KOH pH 7.2, 120 mM K-acetate, 1 mM MgCl$_2$) supplemented after degassing with 1 mM DTT. 190 μl of each solution were pipetted into a 600 μl glass tube, and a 10 μl drop of triolein (>99% purify, T7140 Sigma) was added to the top. In some cases, emulsions were prepared to have a final volume of 100 μl and the drop of triolein was 5 μl. They were vortexed manually at a fixed angle of ~30° for three cycles of 30 s on 30 s off at 25°C under argon atmosphere. Images of resulting emulsions were taken with a compact camera. For analysis by fluorescent microscopy, emulsions were prepared using a mixture of fluorescent and unlabeled protein at a mass ratio 1:20 for Plin4 12mer and 4T > S, and 1:50 for Plin4 4mer.

## Dynamic light scattering (DLS)

Measurements of the mean hydrodynamic radius of the Plin4-oil droplets by dynamic light scattering were performed on a sample taken from the middle of the tube, avoiding any unreacted oil that remained at the top of the emulsion, at least 3 hr after vortexing to prevent the interference of gas bubbles with the measurement. Subsequent samples at later time points were removed in the same manner without any additional mixing. Emulsion samples were diluted 100-fold in freshly degassed

HKM buffer with 1 mM DTT. Measurements were performed on a Zetasizer Nano ZS machine (Malvern) at 25°C, and data were processed using the CONTIN method.

## Observation of protein-oil interaction using microfluidics

Microfluidic experiments were performed in a glass microfluidic chip with a T-junction geometry purchased from Dolomite (part # 3000086 and 3000024). All channels had a rounded cross-section with a 100 µm height and a 110 µm width. Prior to the measurements, the channel walls were wetted with 100 µl of freshly degassed HKM buffer supplemented with 1 mM DTT. The flows were driven and precisely controlled using a piezoelectric pressure control system (OB1 MK3, Elveflow), with typically applied pressures below 300 mbar. After wetting, the main and side inlet channels were filled with buffer and triolein (Sigma), respectively. Injection of oil in the side inlet channel was gently stopped with a manual valve (MV201, LabSmith) before the oil reached the junction (when the meniscus was approximately 400 µm from the junction). In this way, the flow in the main inlet channel could be stopped without significantly affecting the meniscus of oil, and the inlet vial with buffer could be exchanged by a vial with fluorescent protein solution (0.1 mg/ml in HKM buffer, mixed at a ratio 10:1 for unlabeled vs Alexa488-labeled protein). The sample volume in the vial was approximately 400 µl. The flow in the main inlet channel was then resumed and the diffusion of the protein from the main inlet channel into the side channel and its adsorption onto oil meniscus was monitored by time-lapse confocal microscopy for up to 30 min at a rate of 1 frame every 3 s (ECLIPSE TE2000-E, Plan Fluor 40x objective, EZC1 software, Nikon). Finally, the flow in the main inlet channel was stopped again, the inlet vial exchanged by a vial with buffer, and the main inlet channel was rinsed while monitoring the diffusion of the protein from the side channel. For competition experiments between Plin4 12mer and 4T > S mutant, proteins were mixed at a mass ratio 50:1 for unlabeled to labeled protein, Plin4 12mer: Plin4 12mer-Alex488 or 4T > S: Plin4 12mer-Alex488, with a total protein concentration of 0.1 mg/ml. All experiments were conducted at room temperature. Between experiments, glass chips were regenerated by copious washing with 3% SDS at 50°C, followed by distilled water, 3% TFD4 at 50°C, distilled water and finally dried by air (except for the measurements of the surface tension, where inhomogeneous glass surface was needed to increase the contact angle hysteresis). Image analysis was performed with ImageJ/Fiji (*Schindelin et al., 2012*) and Matlab.

## Separation of Plin4-oil emulsion on sucrose gradients

Emulsions were prepared as specified in a final volume of 300 µl including 15 µl of triolein and 0.5 mg/ml of protein. Next, 240 µl of 60% w/v solution of sucrose in HKM buffer with 1 mM DTT was mixed with 240 µl of emulsion, avoiding any oil. A total of 450 µl of this suspension was loaded on the bottom of a centrifuge tube and overlaid with a step sucrose gradient consisting of 300 µl 20%, 300 µl 10%, and 100 µl 0% sucrose in HKM buffer with 1 mM DTT. The samples were centrifuged at 50,000 rpm (214,000 × g) in a Beckman swing-out rotor (TLS 55) for 80 min at 8°C. Four fractions were carefully collected from the bottom with a Hamilton syringe, having the following volumes: 450 µl, 300 µl, 300 µl, and 100 µl, respectively. Equal volumes of all fractions were analysed by protein electrophoresis.

## Circular dichroism

CD measurements were conducted on a Jasco J-815 spectrometer at room temperature with a quartz cell of 0.05 cm path length. For the comparison between wild type and mutant Plin4 in buffer (10 mM Tris, pH 7.5, KCl 150 mM) or in buffer +50% trifluoroethanol (TFE), the spectrum was obtained by averaging six scans recorded from 195 to 260 nm (bandwidth: 1 nm; step size: 0.5 nm; scan speed: 50 nm/min). Ellipiticity was converted to mean residue ellipticity (MRE) by dividing by the product of protein concentration, residue number and path length distance. For the CD spectra of Plin 12mer on oil, emulsions were purified on sucrose gradients in 10 mM Tris, pH 7.5, KCl 150 mM with 0.1 mM DTT and the top 100 µl fraction was collected and placed in the CD cuvette. The spectrum was obtained by averaging 20 scans recorded from 200 to 260 nm (bandwidth: 1 nm; step size: 0.5 nm; scan speed: 20 nm/min). The Plin4 standards were run with these settings. CD spectra were smoothened with the means-movement method with a convolution width of 5.

## Binding of Plin4 to bead-supported bilayers

We prepared bead-supported bilayers using 5 µm silica beads (Bang laboratories) and extruded liposomes composed of 50% diphytanoyl-phosphatidylserine and 50% diphytanoyl-phosphatidylcholine (Avanti Lipids) (Čopič et al., 2018) as described (Pucadyil and Schmid, 2010). Liposomes (200 µM) were incubated with $25 \times 10^6$ beads in 500 µL of HKM buffer for 30 min at room temperature under gentle agitation, and washed three times in HKM buffer with low-speed centrifugation (200 x g for 2 min). To bind fluorescent Plin4 12mer to bead-supported bilayers, freshly-prepared beads were incubated with a mixture of Plin4 12mer and Plin4 12mer-A488 in 30 µL in HKM buffer for 15 min at room temperature, after which the beads were imaged directly by fluorescence microscopy. Alternatively, the fluorescence of the solution before and after incubation was measured in a 96-well opaque plate using a Spectramax M2 fluorimeter (Molecular devices) (excitation 488, emission 525 nm).

## Yeast growth and media

Yeast strains used were: BY4742 MATα *his3Δ1 leu2Δ0 lys2Δ0 ura3Δ0* (Euroscarf), BY4742 *pet10Δ:: KANMX4* (Euroscarf), and BY4741 MATa *his3Δ1 leu2Δ0 met15Δ0 ura3Δ0 PET10-GFP::HisMX* (Huh et al., 2003). Yeast were transformed by standard lithium acetate/polyethylene glycol procedure. Yeast cells expressing different AH constructs were grown in synthetic complete medium lacking uracil (SC-Ura, 6.7 g/l yeast nitrogen base, amino acid supplement without uracil, 2% glucose). To induce LDs, yeast cells either grown in SC-Ura for 24 hr at 30°C (stationary phase) or for 24 hr in SC-Ura, followed by 24 hr incubation in oleic acid (OA) medium (0.67% yeast nitrogen base without amino acids, 0.1% yeast extract, 0.1% (v/v) oleate, 0.25% (v/v) Tween 40, amino acid supplement lacking uracil). For imaging of LDs in early stationary phase, yeast cells were inoculated from a preculture and grown at 30°C in SC-Ura to a final $OD_{600}$ = 1–2.

## Preparation of yeast protein extracts and western blot analysis

Yeast cultures were grown overnight in SD-Ura medium to mid-logarithmic phase. Proteins were extracted from one $OD_{600}$ equivalent of cells by Li-acetate/alkaline extraction as described (Zhang et al., 2011), resuspended in 100 µl of sample buffer (50 mM Tris-HCl, pH 6.8, 100 mM DTT, 2% SDS, 0.1% Bromophenol Blue and 10% glycerol), heated at 65°C for 10 min, and 10 µl was loaded on SDS–PAGE (4–20% Mini-PROTEAN TGX Stain-Free, Bio-Rad). After electrophoresis, total proteins were visualized in the TGX Stain-Free gels (Bio-Rad) after 1 min UV-induced photoactivation with a Gel Doc EZ Imager (Bio-Rad). Proteins were transferred onto a nitrocellulose membrane and perilipin AH-GFP fusions were detected with a rabbit polyclonal anti-GFP antibody (Thermo Fisher Scientific, A11122, 1:5000 dilution). Vps10, when was used as a loading control, was detected with mouse anti-Vps10 monoclonal antibody (Molecular probes, A-21274, 1:100 dilution). Horseradish peroxidase-coupled anti-rabbit (Sigma-Aldrich, A6154) and anti-mouse (GE Healthcare, NA934V) secondary antibodies were used at 1:5000 dilution. Chemiluminescence signals were acquired using Gel Doc EZ Imager.

## Cell culture and transfection

HeLa cells (ATCC) were grown in Dulbecco's modified Eagle's medium (DMEM) supplemented with 4.5 g/l glucose (Life technologies), 10% fetal bovine serum (FBS, Life technology) and 1% Penicillin/ Streptomycin antibiotics (Life technologies). All cell cultures were routinely verified to be free of Mycoplasma by DAPI stain on fixed permeabilized cells. For protein expression, subconfluent cells were transfected with Lipofectamine 2000 (Invitrogen) in Optimem medium (Life technologies) for 6 hr, followed by 16 hr in standard growth medium before the cells were fixed and prepared for imaging.

*Drosophila* S2 cells (ThermoFisher) were cultured in Schneider's *Drosophila* medium (Invitrogen) supplemented with 10% FBS and 1% Penicillin/Streptomycin at 25°C. For generating stably transfected cells, cells were incubated with plasmid DNA and TransIT-Insect Reagent (Mirus), followed by selection with 2 µg/ml puromycin (Life technologies) for 2 weeks. Protein expression from the metal-inducible promoter was induced for 48 hr with the addition of 100 µM Cu-sulfate to the medium. Lipid droplets were induced with 1 mM oleic acid (Sigma) in complex with fatty-acid free BSA (Sigma) for 24 hr. RNAi depletion against CCT1 was performed as described (Čopič et al., 2018).

## Fluorescent microscopy

For imaging of purified protein-oil emulsions, emulsions prepared with fluorescent protein were gently mixed in the glass tube before 1.5 µl of emulsion was withdrawn with a long 200 µl tip and placed on an untreated glass slides (Thermo Scientific). A coverslip was carefully placed on top without applying any pressure. Proteins on bead-supported bilayers were imaged in the same manner.

Yeast cells were harvested by centrifugation, washed, placed on a glass slide and covered with a coverslip. For some experiments, LDs were stained with 1 µg/ml Bodipy 493/503 (Life Technologies) or with Autodot blue dye (Clinisciences) diluted 1000-fol for 30 min at room temp, after which the cells were washed twice and imaged. *Drosophila* S2 cells were imaged on glass slides in the same way as yeast cells.

Transfected HeLa cells were fixed with 3.2% paraformaldehyde (Sigma) in PBS for 30 min at room temp. After washing three times with PBS, cells were stained with Bodipy 493/503 at 1 µg/ml for 30 min at room temperature and washed three times with PBS. Cells were mounted on coverslips with Prolong (Life technologies).

Images of emulsions, bead-supported bilayers, yeast and S2 cells were acquired at room temperature with an Axio Observer Z1 (Zeiss) microscope, equipped with an oil immersion plan-Apochromat 100x/1.4 objective, an sCMOS PRIME 95 (Photometrics) camera, and a spinning-disk confocal system CSU-X1 (Yokogawa) driven by MetaMorph software (Molecular Devices). GFP-tagged or Alex488-labeled proteins and mCherry-tagged or Alex568-labeled proteins were visualized with a GFP Filter 535AF45 and an RFP Filter 590DF35, respectively. When imaging emulsions, images were acquired in 10 to 15 z-sections of 0.2 µm were taken. For imaging HeLa cells and quantification of LD-to-PM signal ratio in yeast, we used an LSM 780 confocal microscope (Zeiss) with a x63/1.4 oil objective and a PMT GaAsP camera, driven by ZEN software. Images were processed with ImageJ and prepared for figures with Canvas Draw (canvas X).

## Fluorescence recovery after photobleaching (FRAP)

FRAP assays in vitro were performed on freshly prepared fluorescent emulsions with Alex488-labeled proteins on glass slides using the CSU-X1 spinning disc microscope and 100x objective, bleaching laser with a wavelength of 473 nm and iLas software controlled by Metamorph. Several circular areas of 25 × 25 pixels were bleached in each field (828 × 960 pixels), either on oil particles or in surrounding solution. The following FRAP time-course was used: six images pre-bleach, then bleach followed by 10 s of 1 image/s, 60 s of 1 image/10 s, and finally 600 s of 1 image/30 s (or until the loss of focus). Fluorescence of the bleached area at each time point was normalized to the average fluorescence before bleaching. Data was processed using Excel.

For FRAP assays in yeast cells, a circular area of 15 × 15 pixels in a cell expressing a GFP-fusion protein was bleached, either on the LDs or on the plasma membrane. Five images were taken before bleaching, followed by a post-bleach time-course: 15 s of 1 image/s, 60 s of 1 image/5 s, and ~200 s of 1 image/20 s. Background fluorescence outside the cell was subtracted from the bleached area and the signal was normalized to the whole cell signal for each time-point. Data was processed with Excel and plotted using SigmaPlot (Systat Software).

FRAP assays in *Drosophila* S2 cells expressing Plin4 12mer-GFP were performed as for yeast, except that three circular areas of 15 × 15 pixels containing isolated LDs were selected per cell. The following FRAP time-course was used: five images pre-bleach, then bleach, followed by 30 s of 1 image/s, 60 s of 1 image/5 s, and finally ~200 s of 1 image/20 s. Data was analyzed in Excel and plotted using SigmaPlot. To obtain the half-time of recovery, average curves from the three FRAP measurements from the same cell were fitted with an exponential-rise equation.

## Protein exchange assay on protein-oil emulsions

Emulsions were prepared as described using unlabeled protein at 0.5 mg/ml and checked by microscopy using CSU-X1 spinning disc microscope (time 0). Then, fluorescent Plin4 12mer-Alexa488 was gently added to the suspension to a final concentration of 0.025 mg/ml (20: 1, unlabeled protein: labeled Plin4 12mer). Samples from the emulsions were withdrawn at indicated time-points without mixing and imaged on glass slides. The re-vortex sample was prepared after 2 hr of incubation by withdrawing 20 µl of the emulsion and vortexing it in a fresh 600 µl glass tube in the same manner as for initial emulsion preparation. Samples were imaged in 10 z-sections of 0.5 µm in

randomly selected fields of 76 µm x 101 µm. The z-section containing the highest number of small droplets was selected for analysis.

## Image analysis

Images were analyzed using ImageJ/Fiji (*Schindelin et al., 2012*). To quantify the number of droplets in protein-oil emulsions, the number of particles in a randomly selected area in a single z-section was counted using 'find maxima' in the fluorescent channel with noise tolerance set to 100. Larger clusters were counted manually. For quantification in the exchange assay, the noise tolerance was set to 150. To quantify Plin4 surface fluorescence on bead-supported bilayers, a circle was drawn on the bead perimeter, converted to a circular line and the mean fluorescence value of the plot profile was calculated. To quantify Plin4 surface fluorescence on oil particles, a line on particles with clear dark middle and the average of the two fluorescence maxima was recorded (the two maxima varied by 6 ± 4%).

To quantify the number of yeast cells with protein signal on LDs, cells were counted manually after applying the same brightness/contrast settings to all images. To quantify the ratio of LD to PM protein signal (mCherry fusions), Pet10-GFP LD marker was used to select the regions of interest (ROIs) corresponding to LDs and the total mCherry fluorescence in the ROIs was recorded. For the quantification of PM fluorescence, images were converted to binary to select the whole yeast perimeter. Then, a band of five pixels was applied to include all of PM signal. After background subtraction, the total LD signal per cell was divided by the total PM signal.

LD size in yeast cells grown in oleic-acid medium was measured using the fluorescent protein. Isolated LDs were fitted manually with a perfect circle and the size of each circular area was recorder. Data were analysed in Excel and plotted with KaleidaGraph (Synergy software).

To determine the fraction of LDs in HeLa cells that were positive for transfected fluorescent protein, a single z-section that contained the most LDs in a cell was first selected. All LDs in the selected cell section were identified in the green (Bodipy dye) channel using the 'Analyze particle' plug-in. LDs positive for fluorescent protein were then identified by determining a threshold value for the red fluorescent signal (mCherry-protein fusion), 1.4x above average cellular fluorescence, and counting all LDs with fluorescence above this threshold. This number was divided by the total LD number to calculate the fraction of LDs in one cell section positive for protein. Data was processed in Microsoft Excel and plotted using SigmaPlot.

## Acknowledgements

We thank C La Torre Garay, M Tuljak, V Countremoulins and N Joly for technical help, and R Schneiter and S Léon for plasmids. We acknowledge the IJM ImagoSeine facility, member of IBiSA and the France-BioImaging infrastructure (ANR-10-INBS-04), and L Bousset, V Albanèse, R Gautier, J Snoj, N Mejhert, C Jackson and Jackson-Verbavatz team for helpful discussions and comments on the manuscript. This work was supported by the CNRS, including a CNRS PICS grant (#214454), the Slovenian Research Agency (research core funding No. P1-0055), and ERC Synergy grant #856404. MG-A was supported by PhD fellowships from the French 'Ministère de l'Education National, de l'Enseignement Supérieur de la Recherche' and Fondation ARC pour la recherche sur le cancer (DOC20190509052).

## Additional information

### Funding

| Funder | Grant reference number | Author |
|---|---|---|
| Centre National de la Recherche Scientifique | CNRS-PICS No 214454 | Alenka Čopič |
| Slovenian Research Agency | No. P1-0055 | Jure Derganc |
| Ministère de l'Enseignement Supérieur et de la Recherche Scientifique | | Manuel Giménez-Andrés |

| Fondation ARC pour la Recherche sur le Cancer | DOC20190509052 | Manuel Giménez-Andrés |
| European Research Council | Synergy #856404 | Bruno Antonny<br>Alenka Čopič |

The funders had no role in study design, data collection and interpretation, or the decision to submit the work for publication.

## Author contributions
Manuel Giménez-Andrés, Conceptualization, Data curation, Formal analysis, Investigation, Methodology, Writing - review and editing; Tadej Emeršič, Conceptualization, Data curation, Formal analysis, Investigation, Methodology; Sandra Antoine-Bally, Data curation, Formal analysis, Investigation, Methodology; Juan Martin D'Ambrosio, Investigation, Writing - review and editing; Bruno Antonny, Conceptualization, Formal analysis, Writing - review and editing; Jure Derganc, Conceptualization, Formal analysis, Supervision, Methodology, Writing - review and editing; Alenka Čopič, Conceptualization, Resources, Data curation, Formal analysis, Supervision, Funding acquisition, Investigation, Visualization, Methodology, Writing - original draft, Project administration, Writing - review and editing

## Author ORCIDs
Manuel Giménez-Andrés  https://orcid.org/0000-0002-4100-306X
Tadej Emeršič  https://orcid.org/0000-0002-6084-5182
Juan Martin D'Ambrosio  https://orcid.org/0000-0003-2834-1838
Bruno Antonny  https://orcid.org/0000-0002-9166-8668
Jure Derganc  https://orcid.org/0000-0001-5135-2231
Alenka Čopič  https://orcid.org/0000-0003-0166-7731

## Decision letter and Author response
Decision letter https://doi.org/10.7554/eLife.61401.sa1
Author response https://doi.org/10.7554/eLife.61401.sa2

# Additional files
## Supplementary files
• Supplementary file 1. Summary of synthetic gene sequences and protein sequences used in this study.

• Supplementary file 2. Calculation of Plin4 12mer density on oil, related to *Figure 8B–D*. Table summarizes input values and calculations for five experimental conditions (A-E) used to standardize the fluorescence of Alexa488 labeled Plin4 12mer on lipid surface, using bead-supported diphytanoyl bilayers as standards.

• Transparent reporting form

## Data availability
All data generated or analysed during this study are included in the manuscript and supporting files. Source data files have been provided for Figures 1 to 8.

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
