## [Decision Letter]

**Acceptance summary:**

Perilipins (PLINs) are ubiquitous lipid droplet proteins that consist of five members in human cells (PLIN1-5). Although all PLIN family members contain N-terminal amphipathic helix (AH) repeat regions that contribute to lipid droplet localization, PLIN4 contains a particularly long AH repeat region as compared to the other members. Here the authors employ a series of complementary biophysics and cell biology approaches to compare the lipid droplet binding properties of the AH repeat regions from various PLIN members. They show that PLIN4, which is the most abundant PLIN in vivo, has the strongest emulsification capacity which is a function of the number of repeats. Moreover, it forms a dense, immobile layer of proteins at the surface of LDs, that stabilize them.

**Decision letter after peer review:**

Thank you for submitting your article "Exceptional stability of a perilipin on lipid droplets depends on its polar residues, suggesting multimeric assembly" for consideration by *eLife*. Your article has been reviewed by 3 peer reviewers, and the evaluation has been overseen by a Reviewing Editor and Vivek Malhotra as the Senior Editor. The reviewers have opted to remain anonymous.

The reviewers have discussed the reviews with one another and the Reviewing Editor has drafted this decision to help you prepare a revised submission.

As the editors have judged that your manuscript is of interest, but, as described below, additional experiments are required before it is published, we would like to draw your attention to changes in our revision policy that we have made in response to COVID-19 (https://elifesciences.org/articles/57162). First, because many researchers have temporarily lost access to the labs, we will give authors as much time as they need to submit revised manuscripts. We are also offering, if you choose, to post the manuscript to bioRxiv (if it is not already there) along with this decision letter and a formal designation that the manuscript is "in revision at *eLife*". Please let us know if you would like to pursue this option. (If your work is more suitable for medRxiv, you will need to post the preprint yourself, as the mechanisms for us to do so are still in development.)

Summary:

Perilipins (PLINs) are ubiquitous lipid droplet proteins that consist of five members in human cells (PLIN1-5). Although all PLIN family members contain N-terminal amphipathic helix (AH) repeat regions that contribute to lipid droplet localization, PLIN4 contains a particularly long AH repeat region as compared to the other members.

Here the authors employ a series of complementary biophysics and cell biology approaches to compare the lipid droplet binding properties of the AH repeat regions from various PLIN members. They show that PLIN4, which is the most abundant PLIN in vivo, has the strongest emulsification capacity which is a function of the number of repeats. Moreover, it forms a dense, immobile layer of proteins at the surface of LDs, that stabilize them.

Essential revisions:

Three referees have reviewed your manuscript. Although there was some concern about the conceptual advance over your previous work, we think that proving in a convincing manner that PLIN4 does form a "coat" at the surface of LD would be a novel result. So far, your conclusion is poorly supported by data and it is not possible to distinguish if PLIN4 does polymerize or forms a frozen/crowded protein layer. We suggest that:

1. You estimate the density of proteins on the LD by calibrating the fluorescence, and test if the proteins are in a close packed state or not.

2. If PLIN4 forms a coat, it could have some structures. It might however be amorphous. Negative staining EM on the LD could distinguish between these 2 types of coats.

3. Phospholipids could affect the way PLIN4 forms a stable structure, and FRAP experiments have been made with pure oil only. Similar in vitro experiments in the presence of artificial lipid droplets generated with both triolein and phospholipids could provide a very different picture. In addition, partial FRAP of a section of droplets in cells and artificial droplets with phospholipids would clarify this point. This should be compared to the other PLIN family members.

[Editors' note: further revisions were suggested prior to acceptance, as described below.]

Thank you for resubmitting your work entitled "Exceptional stability of a perilipin on lipid droplets depends on its polar residues, suggesting multimeric assembly" for further consideration by *eLife*. Your revised article has been reviewed by 2 peer reviewers, and the evaluation has been overseen by a Reviewing Editor and Vivek Malhotra as the Senior Editor. The reviewers have opted to remain anonymous.

Based on the overall interest, we believe the paper merits a revision, but please note that it is the policy of *eLife* to allow only one round of revisions. But given the interest, we will allow you one more revision to address the concerns. We hope that you can address these issues in a timely manner."

The manuscript has been improved but there are some remaining issues that need to be addressed before acceptance, as outlined below: But please note that we will allow only one more revision, so please address the concerns of the reviewers.

The reviewers and myself have considered that the authors have provided useful new data in their revised version and have also included some textual clarifications. Nevertheless, besides showing the unique character of Plin4 as compared to other Perilipins, the major advance of this paper with respect to their 2018 Nature Communications paper would be a convincing demonstration that Plin4 assembles as a coat on LDs in cells. Thus, we think that the authors should provide a more direct proof that Plin4 oligomerizes on LDs, than CD measurements that probe the folding but not the oligomerization, or FRAP on the plasma membrane that is probably not representative of the LD interface. Partial FRAP was performed on LDs in the same *Drosophila* cell type in F. Wilfling et al., Dev. Cell 24, 384 (2013); it is not clear why the authors claim they are too small for this kind of measurements. This type of direct evidence is necessary to unambiguously support the conclusion of your paper.

Reviewer #2:

Several controls (lipid droplets staining, protein levels, helical formation fo purified proteins, etc) are now included. In addition, Figure 8 presents some data suggesting that PLIN4 is present at high concentration in bilayer-covered beads. Although this assay allows only rough estimates, the results provide some support to the model proposed. However, I wonder whether the experiments should also have included some mutants as control, in particular csw mutant which should be defective in step 3 of the model presumably being tested here.

Reviewer #3:

Lipid droplets are specialized organelles that play important roles in the storage of lipids. Perilipins (PLIN1-5) are a ubiquitous family of proteins that are inevitably found on lipid droplets, exhibiting cell-specific expression and unique functions such as signaling integration, lipolytic regulation, and lipid droplet-organelle contacts.

In this manuscript, the authors address whether perilipins exhibit characteristics of protein coats, such as a polymerized structure. The manuscript provides a rigorous comparison of the PLIN proteins and convincingly demonstrates some unique properties of PLIN4, such as its striking stability on lipid droplets due to its elongated amphipathic helix. Mutagenesis identifies important polar residues in PLIN4, raising the possibility that PLIN4 may homo-oligomerize via inter-helical interactions. However, these interactions require biochemical validation and whether PLIN4 forms a "lattice" will require structural and/or high resolution imaging approaches. In addition, whether some of the observed properties extend from model lipid droplets to cellular lipid droplets remains to be examined.

Overall, it is clear that the PLIN4 amphipathic confers unique properties, including a more stable interaction with oil, and that the distribution of the polar residues in PLIN4 is important. This is a modest advance over a previous, very nice paper from the author characterizing PLIN4 and its extended α helix.

In this revised manuscript, the authors provide useful new data (e.g. circular dichroism, LD markers, analysis of expression, etc) and textual clarification of some concerns.

One major concern was the interpretation of experiments examine PLIN association with pure oil droplets, rather than an in vitro lipid droplet that has a phospholipid monolayer. The authors contend that this will be examined in future studies. Thus, it remains a concern that some findings from this artificial system may not translate to PLIN4 properties on cellular lipid droplets.

It was also suggested that partial FRAP be used as one way to determine if the impaired lateral mobility for PLIN4 that is observed in the in vitro system is also present on more complex cellular lipid droplets. The authors mention that this is not possible due to the size of the lipid droplets in *Drosophila* S2 cells and yeast. This is surprising since partial FRAP has been performed on lipid droplets in *Drosophila* S2 cells, such as the examination of GPAT4 mobility (Wilfling et al., Dev Cell 2013). In addition, the PLIN proteins could be expressed in other cell types that do form large lipid droplets. Instead, they examine PLIN mobility on the plasma membrane in pet10∆, which may not accurately reflect their mobility on the monolayer of lipid droplets.

The added data is helpful for addressing some of the prior concerns and the manuscript is a modest advance over the previous 2018 Nature Communications paper.

---

## [Author Response]

Essential revisions:Three referees have reviewed your manuscript. Although there was some concern about the conceptual advance over your previous work, we think that proving in a convincing manner that PLIN4 does form a "coat" at the surface of LD would be a novel result. So far, your conclusion is poorly supported by data and it is not possible to distinguish if PLIN4 does polymerize or forms a frozen/crowded protein layer.

As detailed below, we now provide new evidence for the polymerization model based on (1) protein density measurements, (2) characterization of new mutants, (3) further FRAP analysis, and (4) further measurement of oil particle size. Furthermore, we could refute the amorphous model by showing that Plin4 adopts a helical structure on oil.

We suggest that:1. You estimate the density of proteins on the LD by calibrating the fluorescence, and test if the proteins are in a close packed state or not.

We thank the reviewers for this suggestion. We have performed additional experiments in order to estimate the density of Plin4 AH on oil droplets. In order to standardize the fluorescence of surface-bound Plin4, we developed a new protocol using bead-supported bilayers composed of diphytanoyl phospholipids, to which Plin4 readily binds. These experiments are represented in a new Figure 8 (panels B-D), and described in corresponding sections in the Results and Methods section. Our calculations suggest close packing of Plin4 helices on the surface of oil particles, supporting our model that the helices laterally interact.

2. If PLIN4 forms a coat, it could have some structures. It might however be amorphous. Negative staining EM on the LD could distinguish between these 2 types of coats.

We thank the reviewers for raising this important point. We previously showed that Plin4 AH formed droplets with oil that appeared uniformly stained by negative stain EM (Copic et al., 2018). This is not the case when we form oil droplets using BSA (our unpublished data). Nevertheless, we do not think that negative stain EM provides sufficient resolution to deduce the structure of proteins on the lipid surface,

Instead, we used CD spectroscopy to analyze structure of Plin4 on oil droplets. These new data, which show good agreement between structure of Plin4 12mer in 50% trifluoroethanol and structure of Plin4 12mer on purified oil droplets, are presented in panel A of the new Figure 8. They are also in good agreement with our previously published work, where we showed that purified Plin4 AH constructs (4mer, 12mer and 20mer) formed highly helical structures in 50% TFE, similar to the CD spectrum of Plin4 20mer in the presence of diphytanoyl liposomes (Copic et al., Nat Commun 2018). In all our experiments, Plin4 AH is unfolded in solution (Copic et al., 2018 and this manuscript). As we now explicitly summarize in the model in Figure 8C, Plin4 AH is unfolded in solution and adopts a highly helical structure in contact with a lipid surface. We also demonstrate helix formation for Plin3 AH (Figure 5—figure supplement 1B).

3. Phospholipids could affect the way PLIN4 forms a stable structure, and FRAP experiments have been made with pure oil only. Similar in vitro experiments in the presence of artificial lipid droplets generated with both triolein and phospholipids could provide a very different picture.

We agree with the reviewers that our artificial Plin4-triolein droplets are a highly simplified approximation of LDs. In addition to triolein and Plin4, cellular LDs contain other neutral lipids, phospholipids and other amphipathic lipids, and other proteins. We agree that phospholipids may affect the behavior of Plin4 on LD surface. However, all other components of LDs could also affect the behavior of Plin4. These questions represent crucial directions for our future work.

For now, we compare two extremes: the simplest possible mixture of Plin4 and lipids, and Plin4 on actual LDs in model cells (budding yeast and *Drosophila* S2 cells for the dynamic experiments). We compared the dynamics of Plin4 AH and other Plin AH’s on oil (Figures 1, 2, 5, 7, 8) with their dynamics on LDs (Figures 4, Figure 4 supplement 1, Figure 6). Our results show consistent differences in the relative behavior of Plin4 AH compared to other AHs between in vitro experiments on oil and experiments on LDs in model cells, with Plin4 AH always showing a significantly less dynamic behavior than other Plin AHs or the charge swap mutant. All results presented in this manuscript are consistent with our model that side-chain interactions between polar residues stabilize Plin4 AH on lipid surface,

We entirely agree that phospholipids will likely affect the behavior of Plin4 on LDs. We also expect that other neutral lipids may affect its behavior, or other amphipathic lipids. Our FRAP experiments in *Drosophila* S2 cells (Figure 4 – Supplement 1) in fact suggest that depletion of PC (the main PL species on the LD surface) has a small effect on the mobility of Plin4, whereas another cellular parameter leads to large cell-to-cell differences in the dynamics of Plin4-LD interaction. Due to self-assembly, the binding of Plin4 to LDs should be highly non-linear, and small differences in the starting conditions can lead to large differences in large-scale behavior of Plin4. We will explore this in our future work. In addition to model LDs, where we can more precisely control different parameters, it will be important to test the behavior of Plin4 in a more physiological model, for example in cultured adipocytes, in which Plin4 is endogenously expressed. In addition, please note that we already performed a comprehensive analysis of the binding properties of Plin4 on phospholipid bilayers of different composition (e.g. unsaturation, composition, electrostatic; see Figure 5 in Copic et al., Nat Commun 2018).

Nevertheless, our results, presented in this manuscript, show large differences in the stability of binding to LDs between Plin4 AH and other perilipin AHs. This is the case both on cellular LDs and on protein-triolein particles.

In addition, partial FRAP of a section of droplets in cells and artificial droplets with phospholipids would clarify this point. This should be compared to the other PLIN family members.

Lipid droplets, in particular when covered with Plin4 AH, are too small for partial FRAP (~0.5 μm in diameter in yeast grown in oleic acid, <1 μm in *Drosophila* S2 cells). As an alternative, we performed partial FRAP of Plin4 12mer and Plin1 AH on the plasma membrane of *pet10∆* budding yeast, to which both proteins bind in this yeast mutant. As shown in new panel C in Figure 4, the kinetics of recovery are similar as when we FRAP an entire LD in these cells. These result suggests that, like on the surface of oil, on a phospholipid surface Plin4 AH is also much less mobile than other Plin AHs. We also verified that the differences in mobility of these AHs were not due to differences in their concentration (new Figure 3 – supplement 1B).

[Editors' note: further revisions were suggested prior to acceptance, as described below.]

[…] The reviewers and myself have considered that the authors have provided useful new data in their revised version and have also included some textual clarifications. Nevertheless, besides showing the unique character of Plin4 as compared to other Perilipins, the major advance of this paper with respect to their 2018 Nature Communications paper would be a convincing demonstration that Plin4 assembles as a coat on LDs in cells. Thus, we think that the authors should provide a more direct proof that Plin4 oligomerizes on LDs, than CD measurements that probe the folding but not the oligomerization, or FRAP on the plasma membrane that is probably not representative of the LD interface. Partial FRAP was performed on LDs in the same *Drosophila* cell type in F. Wilfling et al., Dev. Cell 24, 384 (2013); it is not clear why the authors claim they are too small for this kind of measurements. This type of direct evidence is necessary to unambiguously support the conclusion of your paper.

We are glad that the editors and the reviewers find that our extensive revisions improved our manuscript.

We addressed the remaining concern by providing further evidence that Plin4 AH assembles as an LD coat in cells, in several ways:

1. We now provide additional FRAP of Plin4-derived AHs on the surface of LDs in HeLa cells. This data is presented in new panel H of Figure 6, replacing the previous FRAP experiment with the csw mutant that was done on the plasma membrane in yeast. A single LD or small LD cluster per cell was bleached (see Author response image 1); however, like in other systems, the small size of these LDs did not permit us to perform partial FRAP. Again, in Hela cells, we noticed a dramatic effect of modifying the distribution of the charge residues of the protein (charge swap mutant), which is in perfect line with what we observed in yeast and S2 cells. These experiments were conducted with a short 4mer form of Plin4 AH, which is made slightly more hydrophobic (2T to V substitutions per 33aa repeat) in order to stain all lipid droplets, hence facilitating the analysis. The control construct displayed slow recovery (time constant in the range of 10 min). When we reorganized the charged residues, the fluorescence recovery became at least 60-times faster. These experiments indicate that charged residues strongly influence the stability of Plin4 AH on LDs in HeLa cells and that the contribution of hydrophobic interaction is less decisive than electrostatics. This additional experiment also addresses the concern of reviewers about the validity of our experiments performed at the PM, which we agree is not the same as an LD surface.

**Author response image 1. sa2fig1:** Image showing a HeLa cell expressing Plin4 AH before FRAP. The area that was bleached is marked.

2. We have provided additional evidence of coat-like properties of Plin4 AH (see previous supplementary figure to Figure 4, panels A-C). This data may have been overlooked by the reviewers.

This figure shows that while the dynamics of Plin4 AH can vary between cells, it correlates with protein concentration on LDs. Correlation between protein concentration and stability of binding is a defining feature of coat-like proteins that polymerize into a lattice, as has been reported in the literature (e.g., see Saleem et al. 2015, or Sorre et al., 2012, referenced in the manuscript). It is the opposite of what you would expect in the case of protein crowding, which has been suggested to be important for determining protein composition of LDs (see Kory et al., 2015, referenced in the manuscript).

Furthermore, this figure also shows the experiment that reviewers would like us to add, i.e. FRAP of Plin4 AH on LDs in S2 cells – except that, as we already explained in our first rebuttal letter, these LDs were too small to permit doing a partial FRAP. In contrast to the LDs that were partially bleached by Wilfling at al, which measured ~4 μm in diameter, the Plin4-covered LDs in S2 cells measure ~1μm in diameter (please see Figure 7D in Copic et al., 2018 for a thorough quantification). This difference is due to two reasons: as shown by Wilfling et al., LDs harboring GPAT4 grow; and, as shown by us in this present study, Plin4 AH reduces the size of LDs even under wild-type conditions. Therefore, the two experiments are simply not comparable, even though they were performed in the same cell line.

3. We would like to emphasize that the differences in dynamics of FRAP that we observe between Plin4 AH and Plin1/2/3 AH on LDs in yeast, or between Plin4 mutants in HeLa cells, are at the scale of about 2 orders of magnitude, and that we are studying peripheral membrane proteins that clearly bind to LDs from the cytosol. In this context, we would not expect that bleaching a full LD or a part of an LD could give significantly different results.

4. It has been suggested in the literature that ‘S3-12 [i.e., Plin4] coats nascent LDs’. This is in fact the title of the paper published by Wolins et al., in 2003 and this statement has been repeated in many reviews. However, this suggestion is based on some observations in fixed cultured adipocyte cell line, without any further experimental evidence. Therefore, we agree that the suggestion that Plin4 coats nascent LDs is not new. However, we take here an entirely different approach from Wolins et al., which permitted us to test more rigorously and extensively the model. Please note that the size of these ‘nascent LDs’ in cultured adipocyte cells is also too small for partial FRAP. Moreover, it is not clear to what extent these cells actually reproduce the characteristics of LDs in adipocyte tissue, i.e., where an adipocyte often contains a single giant LD.

Finally, we would like to emphasize that we have actually not invented the term ‘LD coat’ to describe perilipins; this term has been quite often used in the literature, although it has until now never been formally investigated.

It would be preposterous to expect that a single study could unequivocally resolve the question what it means to be an LD coat. Indeed, we discuss this point extensively in the Discussion and point out many unresolved questions. We believe that our study sets the debate on a formal ground and suggests further testable hypotheses, thus representing an important step towards understanding the function of perilipins and the behavior of LD proteins in general.

Reviewer #2:Several controls (lipid droplets staining, protein levels, helical formation fo purified proteins, etc) are now included. In addition, Figure 8 presents some data suggesting that PLIN4 is present at high concentration in bilayer-covered beads. Although this assay allows only rough estimates, the results provide some support to the model proposed. However, I wonder whether the experiments should also have included some mutants as control, in particular csw mutant which should be defective in step 3 of the model presumably being tested here.

First, this requirement was not stated in the first round of reviewers’ comments. Second, the new FRAP data shown in Figure 6H, together with other approaches, clearly shows that the CSW mutant is much more dynamic at the LD surface. Third, the estimation of the absolute level of Plin4 12mer density on oil surface required the development of a new model system to relate surface density of Plin4 AH to its surface fluorescence (i.e., super beads coated with diphytanoyl liposomes) and required careful calibration and optimization. It represented the major part of our revisions. It also required careful analysis of the fluorescently-labelled protein, to make sure that the fluorescent moiety did not affect the binding properties of the AH. For the latter reason, we favored competition experiments with unlabeled proteins over comparing many different labeled proteins. Whereas doing additional experiments to measure the density of other proteins/constructs/mutants would certainly be feasible, it would extend our experimental work way over the time that *eLife* generally suggests for revisions, and would also require the first author to postpone his postdoctoral training.

Reviewer #3:Lipid droplets are specialized organelles that play important roles in the storage of lipids. Perilipins (PLIN1-5) are a ubiquitous family of proteins that are inevitably found on lipid droplets, exhibiting cell-specific expression and unique functions such as signaling integration, lipolytic regulation, and lipid droplet-organelle contacts.In this manuscript, the authors address whether perilipins exhibit characteristics of protein coats, such as a polymerized structure. The manuscript provides a rigorous comparison of the PLIN proteins and convincingly demonstrates some unique properties of PLIN4, such as its striking stability on lipid droplets due to its elongated amphipathic helix. Mutagenesis identifies important polar residues in PLIN4, raising the possibility that PLIN4 may homo-oligomerize via inter-helical interactions. However, these interactions require biochemical validation and whether PLIN4 forms a "lattice" will require structural and/or high resolution imaging approaches. In addition, whether some of the observed properties extend from model lipid droplets to cellular lipid droplets remains to be examined.

We fully agree that structural studies will be required to test our lattice model. Given the predicted size of Plin4 AH, between 0.5 and 1 nm in diameter, and the high density of AHs required for our lattice model (and observed in our in vitro experiment), these structural and high-resolution imaging approaches will be far from straight-forward to implement. Please note that the exact molecular arrangement of apolipoproteins on lipid particles is not clear despite decades of research on this topic!

Overall, it is clear that the PLIN4 amphipathic confers unique properties, including a more stable interaction with oil, and that the distribution of the polar residues in PLIN4 is important. This is a modest advance over a previous, very nice paper from the author characterizing PLIN4 and its extended α helix.

We thank the reviewer for the kind comment about our previous paper. We would like to note in this context that the main message of our previous paper has been hardly noted within the LD community. Specifically, we show by using the abundant LD protein Plin4 as a model, along with a set of mutants where we carefully modulate one by one defining parameters of an AH (length, hydrophobicity, charge), that even AHs with a very low hydrophobicity and completely lacking any bulky hydrophobic residues can efficiently target LDs. Whereas increasing the hydrophobicity of an AH does improve binding to LDs, it also makes it more promiscuous, as such AHs then start to invade other cellular compartments. According to our results with Plin4 AH, which are in the present work confirmed by a careful comparison with other perilipins, low hydrophobicity in fact improves specificity of targeting to LDs. Whereas the efficiency of LD targeting of Plin4 may be due to the specific organization of its charged residues, we show in the present work that this is not the case for other Plin AHs. In addition to being poorly hydrophobic, all these AHs are relatively very long, thereby using the increased size of their lipid binding surface to target LDs.

At the same time as our paper was published, the study of Prevost et al., 2018, suggested the opposite mechanism, i.e. that bulky hydrophobic residues were required for AH targeting to LDs. This contrasting conclusion has been almost unequivocally adopted within the scientific community. In more than a few cases, our work is even cited next to the work of Prevost et al. as supporting evidence that bulky hydrophobic residues are needed to target LDs. We try to rectify this oversight in our present work, and we hope that this example illustrates the relevance of our continuing studies of perilipin AHs. Moreover, we are not aware of any equivalent effort to understand the mechanism of LD coating by perilipins. Our manuscript includes (i) multiple kinetics approaches, both in vivo in several cell lines and in vitro with several unique setups (microfluidic, exchange assays, FRAP); (ii) CD spectroscopy on oil (which has never been performed); (iii) a comprehensive mutagenesis analysis with subtle mutations like N>Q, K>R, charge swapped that have dramatic effects on perilipin targeting.

In this revised manuscript, the authors provide useful new data (e.g. circular dichroism, LD markers, analysis of expression, etc) and textual clarification of some concerns.One major concern was the interpretation of experiments examine PLIN association with pure oil droplets, rather than an in vitro lipid droplet that has a phospholipid monolayer. The authors contend that this will be examined in future studies. Thus, it remains a concern that some findings from this artificial system may not translate to PLIN4 properties on cellular lipid droplets.

The relevant LDs are the ones found in cells that express endogenous Plin4, like adipocytes. The surface area of LDs in mature adipocytes is huge. The protein concentration of Plin4 in mature adipocytes has to our knowledge not been measured; however, we would not expect Plin4 to be present at a concentration that would allow it to cover the total surface of LDs, but rather to form patches. We hope that the reviewer can appreciate that addressing this hypothesis will be technically extremely demanding. Despite the importance of understanding adipocyte function for human health, and the fact that Plin4 is a highly abundant adipocyte protein, since the study of Wolins et al., published in 2003, the coating function of Plin4 in adipocyte-like cells has not been further addressed (please see also our response to the editors above). In the future, it will be important to study Plin4 in an authentic adipocyte, but this is a highly demanding system. What makes the present work relevant is the strong correlation between observations with pure oil in vitro and cellular observations in disparate cell systems (yeast, insect and mammalian cells) and for many Plins (Plin1, 2, 3, 4 and mutants).

We would also like to add that there is likely a great variability in the properties of an LD surface, compared to any other organelle. Unlike in a bilayer, the packing density of an LD monolayer can fluctuate to a high extent, thus changing LD surface tension; this will happen during LD growth, shrinkage, fusion, vesicular budding from the surface, formation of ER-LD bridges, action of lipid enzymes. It is also not clear at all to what extent the neutral lipids from the LD core intercalate into the surface monolayer. All these parameters are expected to vary during the lifetime of a cell, and also between different cells, making the choice of an in vitro mimetic system very challenging.

It was also suggested that partial FRAP be used as one way to determine if the impaired lateral mobility for PLIN4 that is observed in the in vitro system is also present on more complex cellular lipid droplets. The authors mention that this is not possible due to the size of the lipid droplets in *Drosophila* S2 cells and yeast. This is surprising since partial FRAP has been performed on lipid droplets in *Drosophila* S2 cells, such as the examination of GPAT4 mobility (Wilfling et al., Dev Cell 2013). In addition, the PLIN proteins could be expressed in other cell types that do form large lipid droplets. Instead, they examine PLIN mobility on the plasma membrane in pet10∆, which may not accurately reflect their mobility on the monolayer of lipid droplets.

We agree that performing FRAP experiments on the plasma membrane provided only indirect indications on the behavior of Plins at the LD surface. Therefore, we now present new FRAP experiments on authentic LDs (see new Figure 6H). Please note, however, that we cannot perform partial FRAP on lipid droplets in *Drosophila* S2 cells. This is because Plin4 reduces the size of LD (≈ 1 µm), whereas the expression of GPAT4 by Wilfling et al., resulted in much larger (4 µm) LDs.

The added data is helpful for addressing some of the prior concerns and the manuscript is a modest advance over the previous 2018 Nature Communications paper.

We have also revised the abstract to highlight better the general importance of this work. We agree with the reviewer that the previous version of the abstract was not sufficiently strong.